# Newton Method Revisited: Global Convergence Rates up to $\mathcal{O}\left(k^{-3}\right)$ for Stepsize Schedules and Linesearch Procedures

**Slavomír Hanzely**
CISPA Helmholtz Center for Information Security, Saarbrücken, Germany
Mohammed bin Zayed University of Artificial Intelligence, Abu Dhabi, UAE
shanzely@gmail.com

**Farshed Abdukhakimov & Martin Takáč**
Mohammed bin Zayed University of Artificial Intelligence, Abu Dhabi, UAE
{farshed.abdukhakimov,martin.takac}@mbzuai.ac.ae

## Abstract

This paper investigates the global convergence of stepsized Newton methods for convex functions with Hölder continuous Hessians or third derivatives. We propose several simple stepsize schedules with fast global convergence guarantees, up to $\mathcal{O}\left(k^{-3}\right)$. For cases with multiple plausible smoothness parameterizations or an unknown smoothness constant, we introduce a stepsize linesearch and a backtracking procedure with provable convergence as if the optimal smoothness parameters were known in advance. Additionally, we present strong convergence guarantees for the practically popular Newton method with exact linesearch.

## 1 Introduction

Second-order methods are fundamental to scientific computing. With its rich history that can be traced back to works Newton (1687), Raphson (1697), (Simpson, 1740), they have remained widely used up to the present day (Ypma, 1995; Conn et al., 2000). The main advantage of second-order methods is their independence from the conditioning of the underlying problem, enabling an extremely fast local quadratic convergence rate, where precision doubles with each iteration. Additionally, they are inherently invariant to rescaling and coordinate transformations, which greatly simplifies their implementation. In contrast, the convergence of first-order methods is highly dependent on the problem's conditioning, resulting in a slower linear convergence rate and a greater sensitivity to parameter choice. Despite their natural geometry and extremely fast local convergence, second-order methods often lack global convergence guarantees. Even the classical Newton method,

$$x^{k+1} = x^k - \left[\nabla^2 f(x^k)\right]^{-1} \nabla f(x^k), \tag{1}$$

can diverge when initialized far from the solution (Jarre & Toint, 2016; Mascarenhas, 2007). Global convergence guarantees are typically achieved through various combinations of stepsize schedules (Nesterov & Nemirovski, 1994), line search procedures (Kantorovich, 1948; Nocedal & Wright, 1999), trust-region methods (Conn et al., 2000), and Levenberg-Marquardt regularization (Levenberg, 1944; Marquardt, 1963). The simplest globalization strategy is to employ stepsize schedules $\alpha_k$,

$$x^{k+1} = x^k - \alpha_k \left[\nabla^2 f(x^k)\right]^{-1} \nabla f(x^k), \tag{2}$$

often based on implicit descent conditions, requiring an additional subroutine per iteration, such as exact linesearch (Cauchy, 1847; Shea & Schmidt, 2025), Armijo linesearch (Armijo, 1966), Wolfe condition (Wolfe, 1969), Goldstein condition (Nocedal & Wright, 1999). However, those methods often lack global convergence guarantees achieved by simple stepsize schedules. Notably, Nesterov & Nemirovski (1994) introduced a damped stepsize schedules with global rate $\mathcal{O}\left(k^{-\frac{1}{2}}\right)$. Hanzely et al. (2022) improved this result by discovering duality between Lavenberg-Marquardt regularization and

Newton stepsizes and proposing a stepsize with global rate $\mathcal{O}\left(k^{-2}\right)$ matching regularized Newton methods (Nesterov & Polyak, 2006; Mishchenko, 2023; Doikov & Nesterov, 2024).

Despite recent advances, existing guarantees still fall short of the optimal rate $\Omega\left(k^{-\frac{7}{2}}\right)$ for functions with Hölder-continuous Hessians (Gasnikov et al., 2019; Agarwal & Hazan, 2018; Arjevani et al., 2019), leaving open the question of whether more efficient step-size schedules remain to be discovered.

In the context of first-order methods, several nontrivial step-size schedules are known to improve the convergence of Gradient Descent. Young (1953) introduced a Chebyshev polynomial–based schedule that attains the optimal rate for quadratic objectives. Polyak (1987) proposed a schedule that is optimal for nonsmooth convex functions, and Altschuler & Parrilo (2023); Grimmer et al. (2025) developed schedules achieving semi-accelerated rates for general convex, Lipschitz-smooth objectives. These results naturally prompt the question of whether improved stepsize schedules for Newton's method can be found.

We answer this positively. We show that a stepsized Newton method can be analyzed under an alternative assumption – Hölder continuity of the third derivatives – yielding convergence guarantees reminiscent of third-order tensor methods, up to $\mathcal{O}(k^{-3})$[1]. Analyzing Newton's method through assumptions on third derivatives is, to the best of our knowledge, a novel and somewhat unexpected perspective, given that Newton's method is typically viewed as the canonical second-order method.

## 1.1 BENEFITS OF SIMPLE METHODS

While it is possible to achieve optimal rates using acceleration techniques with a more complex structure (Gasnikov et al., 2019), simple methods are often preferred in practice for several reasons.

Firstly, they are simple and easy to understand. They are also inherently robust, typically involving fewer hyperparameters, which minimizes the need of hyperparameter tuning. In contrast, accelerated methods often require multiple sequences of iterates and additional hyperparameters, significantly increasing the complexity of tuning.

Moreover, basic methods can be seamlessly integrated with various techniques to enhance practical performance, such as parameter searches, data sampling strategies, momentum estimation, and gradient clipping. Combining these techniques with accelerated methods, however, introduces significant challenges. In the context of first-order methods, acceleration with parameter searches provides limited improvement over Gradient Descent with stepsize linesearch (Shea & Schmidt, 2024; Fox & Schmidt, 2024).

For second-order methods, the stepsized Newton method is popular due to its affine invariance (i.e., invariance to changes in basis and scaling), making it an efficient and convenient optimization tool.

## 1.2 NOTATION

For convex function $f : \mathbb{R}^d \to \mathbb{R}$, we consider the optimization objective

$$\min_{x \in \mathbb{R}^d} f(x), \tag{3}$$

where $f$ is twice differentiable with nondegenerate Hessians that is potentially ill-conditioned. We denote any minimizer of the function as $x^* \in \operatorname{argmin}_{x \in \mathbb{R}^d} f(x)$ and the optimal value $f_* \stackrel{\text{def}}{=} f(x^*)$. We define norms based on a symmetric positive definite matrix $\mathbf{H} \in \mathbb{R}^{d \times d}$. For all $x, g \in \mathbb{R}^d$,

$$\|x\|_{\mathbf{H}} \stackrel{\text{def}}{=} \langle \mathbf{H}x, x \rangle^{1/2}, \quad \|g\|_{\mathbf{H}}^* \stackrel{\text{def}}{=} \langle g, \mathbf{H}^{-1}g \rangle^{1/2}.$$

As a special case $\mathbf{H} = \mathbf{I}$, we get $l_2$ norm $\|x\|_{\mathbf{I}} = \langle x, x \rangle^{1/2}$. We will be utilizing *local Hessian norm* $\mathbf{H} = \nabla^2 f(x)$, with a shorthand notation $\|h\|_x \stackrel{\text{def}}{=} \|h\|_{\nabla^2 f(x)}, \|g\|_x^* \stackrel{\text{def}}{=} \|g\|_{\nabla^2 f(x)}^*$ for $h, g \in \mathbb{R}^d$.

---

[1] Under Hölder continuity of third derivatives, the attainable lower bound is $\Omega\left(k^{-5}\right)$ (Gasnikov et al., 2019).

For the Hessians and third derivatives we will be measuring them in an operator norm. Given the iterate $x$, operator norm of matrix $\mathbf{H}$ and three dimensional tensor $\mathbf{T}$ are defined as

$$\|\mathbf{H}\|_{op} \stackrel{\text{def}}{=} \sup_{y \in \mathbb{R}^d} \frac{\|\mathbf{H}y\|_x^*}{\|y\|_x}, \qquad \|\mathbf{T}\|_{op} \stackrel{\text{def}}{=} \sup_{y,z,w \in \mathbb{R}^d} \frac{|\mathbf{T}[y,z,w]|}{\|y\|_x\|z\|_x\|w\|_x}.$$

In this work, we use these operator norms exclusively with $x = x^k$ and $y = z = w = x^{k+1} - x^k$.

## 1.3 Stepsizes as a form of regularization

Hanzely et al. (2022) demonstrated that a stepsize schedule for the Newton method is equivalent to cubical regularization of the Newton method (Nesterov & Polyak, 2006) if the regularization is measured in the local Hessian norms. As the regularized Newton methods leverage the Taylor polynomial, we denote the second-order Taylor approximation of $f(y)$ by information at point $x$ as

$$\Phi_x(y) \stackrel{\text{def}}{=} f(x) + \langle \nabla f(x), y - x \rangle + \frac{1}{2}\|y - x\|_x^2.$$

In particular, Hanzely et al. (2022) showed that

$$x^{k+1} = T(x^k), \qquad T(x) \stackrel{\text{def}}{=} \operatorname*{argmin}_{y \in \mathbb{R}^d} \left\{ \Phi_x(y) + \frac{\sigma}{3}\|y - x\|_x^3 \right\}$$

is equivalent to a Newton method with stepsize AICN[2]

$$x^{k+1} = x^k - \alpha_k [\nabla^2 f(x^k)]^{-1} \nabla f(x^k), \qquad \text{for } \alpha_k = \frac{2}{1 + \sqrt{1 + 2\sigma\|\nabla f(x^k)\|_{x^k}^*}}. \tag{4}$$

Note that stepsize schedule (4) preserves much larger stepsize when initialized far from the solution, $\|\nabla f(x^0)\|_{x^0}^* \gg 1$, compared to the stepsize of Damped Newton method (Nesterov & Nemirovski, 1994), which sets stepsize for $L_{sc}$-self-concordant functions as $\alpha_k = \frac{1}{1 + L_{sc}\|\nabla f(x^k)\|_{x^k}^*}$. Aiming to extend these results beyond self-concordance, in Section 2 we present algorithm RN that under much more general $L_{p,\nu}$-Hölder continuity (Definition 1) and $q = p + \nu \in [2, 4]$ supports stepsize $\alpha_k = \frac{1}{1 + (9L_{p,\nu})^{\frac{1}{q-1}}\|\nabla f(x^k)\|_{x^k}^{*\frac{q-2}{q-1}}}$, matching AICN's asymptotic dependence on gradient norm and smoothness constant (for $L_{2,1}$-Hölder continuous functions, $q = 3$) and constant stepsizes of Karimireddy et al. (2018); Gower et al. (2019a) (for $L_{2,0}$-Hölder continuous functions, $q = 2$).

**Remark.** *Stepsized Newton methods often enjoy much simpler analysis compared to Newton methods regularized in $l_2$ norms, as they can seamlessly transition between gradients and model differences,*

$$\left\|x^{k+1} - x^k\right\|_{x^k} \stackrel{(4)}{=} \alpha_k \left\|\nabla f(x^k)\right\|_{x^k}^*. \tag{5}$$

## 1.4 Higher order of regularization

Extending cubic regularization (Nesterov & Polyak, 2006), tensor methods achieve better convergence guarantees by regularizing $p$-th order Taylor approximations by $(p + 1)$-th order regularization (see survey Kamzolov et al. (2023)). In particular, for third-order tensor methods, Nesterov (2021) showed that regularization can avoid computation of third-order derivatives, and Doikov et al. (2024) simplified this regularization using technique of Mishchenko (2023) to

$$x^{k+1} = T(x^k), \qquad \text{where } T(x) = \operatorname*{argmin}_{y \in \mathbb{R}^d} \left\{ \Phi_x(y) + \frac{\sigma}{2}\|y - x\|_2^2\|\nabla f(x)\|_2^\beta \right\}, \tag{6}$$

for $\beta, \sigma \geq 0$. Combining insights about higher-order regularization with the regularization-stepsize duality of Hanzely et al. (2022), we show that the higher-order regularization in local norms

$$x^{k+1} = T_{\sigma,\beta}(x^k), \qquad \text{where } T_{\sigma,\beta}(x) = \operatorname*{argmin}_{y \in \mathbb{R}^d} \left\{ \Phi_x(y) + \frac{\sigma}{2 + \beta}\|y - x\|_x^{2+\beta} \right\}, \tag{7}$$

---

[2]Hanzely et al. (2022) expressed the stepsize as $\alpha_k = \frac{-1 + \sqrt{1 + 2\sigma\|\nabla f(x^k)\|_{x^k}^*}}{\sigma\|\nabla f(x^k)\|_{x^k}^*}$, we simplified this form.

Table 1: Global convergence guarantees of stepsized Newton methods under various notions of Hölder continuity (Definition 1). For simplicity, we report dependence only on the number of iterations $k$.

| Stepsize schedule | Stepsize for $g_x \overset{\text{def}}{=} \|\nabla f(x)\|_x^*$ | Smoothness assumption | Global rate | Reference |
|---|---|---|---|---|
| Damped Newton B | $\frac{1}{1+L_{sc}g_x}$ [0] | $L_{sc}$ [0] | $\mathcal{O}\left(k^{-\frac{1}{2}}\right)$ [1] | (Nesterov & Nemirovski, 1994) [1] |
| AICN | $\frac{2}{1+\sqrt{1+2L_{2,1}g_x}}$ [2] | $L_{2,1}$ | $\mathcal{O}\left(k^{-2}\right)$ | (Hanzely et al., 2022) |
| RN (Algorithm 1) | $\frac{1}{1+(9L_{p,\nu})^{\frac{1}{q-1}}g_x^{\frac{q-2}{q-1}}}$ [3] | $L_{p,\nu}$ [3] | $\mathcal{O}\left(k^{-(p+\nu-1)}\right)$ [3] | **This work** (Theorem 4) |
| GRLS (16) | Linesearched | $L_{p,\nu}$ [3] (unknown) | $\min_{p,\nu}\mathcal{O}\left(k^{-(p+\nu-1)}\right)$ [3] | **This work** (Corollary 1) |
| UN (Algorithm 2) | Backtracked | $L_{p,\nu}$ [3] (unknown) | $\min_{p,\nu}\mathcal{O}\left(k^{-(p+\nu-1)}\right)$ [3] | **This work** (Theorem 5) |
| Greedy Newton (18) | Linesearched | $L_{p,\nu}$ [3] (unknown) | $\min_{p,\nu}\mathcal{O}\left(k^{-(p+\nu-1)}\right)$ [3] | Folklore Rate: Corollary 2 **(new)** |

[0] Constant $L_{sc}$ represents self-concordance constant and is implied by $L_{2,1}$-Hölder continuity.

[1] Authors show global decrease $f(x^{k+1}) \leq f(x^k) - c$ for some $c > 0$. Rate $\mathcal{O}(k^{-\frac{1}{2}})$ is reported in Hanzely et al. (2022), but we were unable to find or prove or the rate for Damped Newton B of the form $\mathcal{O}(k^{-\alpha})$.

[2] Authors expressed the stepsize as $\frac{-1+\sqrt{1+L_{2,1}g_x}}{L_{2,1}g_x}$, we present a simplified equivalent form.

[3] Parameters $p, \nu$ are fixed and satisfy $p \in \{2, 3\}$, $\nu \in [0, 1]$ and $p + \nu - 1 \in [1, 3]$.

is equivalent to a Newton method with stepsize $\alpha_k \in (0, 1]$, and $\alpha_k$ is the *unique* positive root of the polynomial $P[\alpha] \overset{\text{def}}{=} 1 - \alpha - \alpha^{1+\beta}\sigma\|\nabla f(x^k)\|_{x^k}^{*\beta}$. Even though the polynomial $P$ lacks an explicit formula for its roots, we derive algorithm RN with a simple and exactly computed stepsize.

This method can leverage similarity of the third-derivatives similarly to Nesterov (2021, Lemma 3).

**Lemma 1.** *Let function $f : \mathbb{R}^d \to \mathbb{R}$ be third-order $L_{3,\nu}$-Hölder continuous (Definition 1). Then*

$$\left\|\nabla^3 f(x^k)[x^{k+1} - x^k]^2\right\|_{x^k}^* \leq 2\left(\frac{L_{3,\nu}}{1+\nu}\right)^{\frac{1}{1+\nu}}\left\|x^{k+1} - x^k\right\|_{x^k}^2 \qquad \forall x^k, x^{k+1} \in \mathbb{R}^d.$$

Notably, formulation (7) is very general, and it also encapsulates all polynomial upper bounds of polynomials $P[\|x - y\|_x]$ with smaller exponents. We refer the reader for more details to Section F.

## 1.5 CONTRIBUTIONS

We summarize our contributions below, with detailed comparison to the most relevant literature discussed in Section 1.6.

- **Newton method under third-order tensor similarity:**
  We analyze the stepsized Newton method for functions with Hölder continuous third-derivatives (Definition 1), connecting the classical second-order Newton method to third-order tensor methods.

- **Simple stepsizes for fast global convergence:**
  We propose multiple stepsize schedules for the Newton method (RN, Algorithm 1), leveraging **various** Hölder continuity assumptions (Definition 1). Although the stepsize is chosen to be a root of a non-quadratic polynomial, it is surprisingly **simple and directly computable**.

  Depending on the considered variant of the Hölder continuity assumption, they achieve a global convergence rate up to $\mathcal{O}\left(k^{-3}\right)$ (Theorem 2). These are the first Newton method stepsizes improving upon the rate $\mathcal{O}\left(k^{-2}\right)$ of Hanzely et al. (2022). Additionally, we establish the following rates:

  – a **local superlinear** convergence rate (Theorem 3),

- – a **global linear** convergence (Theorems 8, 9) under additional assumption of finite *s-relative size* (Definition 2) (Doikov et al., 2024),
- – and a **global superlinear** convergence (Theorem 7) under the additional assumption of uniform star-convexity (Definition 3) of degree $s \geq 2$.

- **Stepsize linesearches for unknown parameters:**
  In practice, smoothness constants are often unknown, requiring approximation or fine-tuning. To address this, we introduce a theoretical **linesearch** procedure GRLS (16) and a practical **stepsize backtracking** method UN (Algorithm 2), both of which provably converge as if the **optimal** parameterization was known in advance (Corollary 1, Theorem 5).

- **Guarantees for popular Newton linesearch:**
  As a byproduct of our analysis, we obtain convergence guarantees for the popular Newton method with greedy linesearch (18) (Corollary 2, Theorem 7). This is, to our best knowledge, the first such result.

- **Experimental comparison:**
  In Section 5, we compare the proposed algorithms (RN, UN, and GRLS) with existing methods and demonstrate that they outperform their counterparts in most of the considered scenarios. Also, we show that the linesearch procedure GRLS resemble stepsizes of popular Greedy Newton linesearch.

### 1.6 DETAILED COMPARISON TO THE MOST RELEVANT LITERATURE

Our theoretical framework builds on several insights from Hanzely et al. (2022) and Doikov et al. (2024). We now outline the key differences between these approaches and ours.

Compared to our approach, the AICN method of Hanzely et al. (2022) is restricted to cubic regularization and achieves only an $\mathcal{O}\left(k^{-2}\right)$ convergence rate. In contrast, our schedules accommodate a broader range of smoothness notions, including Hölder continuity of the third derivative, enabling Algorithm 1 to achieve rates up to $\mathcal{O}\left(k^{-3}\right)$. Moreover, while AICN requires prior knowledge of the smoothness constant, our backtracking linesearch Algorithm 2 provably converges as if the optimal parametrization were known in advance.

Furthermore, while cubic regularization in Hanzely et al. (2022) lead to the stepsize defined as the root of a quadratic polynomial, higher-order regularizations require a stepsize given by a root of a higher-order polynomial. Surprisingly, we show that even with higher-order regularization there exists a unique positive root in the interval $(0, 1]$, and we propose algorithms (Algorithm 1 and Algorithm 2) that can operate without any additional linesearch.

In comparison to Doikov et al. (2024), which utilizes standard $l_2$ norms for regularization, our approach employs the local Hessian norms suggested by Hanzely et al. (2022). With local norms, the minimizers of the various regularization models (7) lie on the same line. This provides a natural geometric connection between different regularizations, and can be further formalized by the notion of loss-transformation invariance (Shestakov et al., 2025). Local norms also yield a simpler algorithm that is invariant under linear transformations (e.g., data scaling or change of basis), a highly practical property that reduces hyperparameter tuning.

From a technical point of view, although our proofs draw on techniques from Doikov et al. (2024), they cannot be directly adapted to the setting of local norms. The main difficulty is that the stepsize $\alpha_k$ appears raised to the power $1 + \beta$, which propagates nontrivially throughout the analysis and complicates adaptation. Our key insight is a reparametrization (line 141) in which a single implicit parameter $\theta$ encapsulates both $\beta$ and $\sigma$. This reparametrization allows us to recover a proof structure similar to that of Doikov et al. (2024) while avoiding direct manipulations of term $\alpha_k^{1+\beta}$.

We also emphasize that our results provide a theoretical explanation for the success of popular stepsize linesearch rules along the Newton direction. These insights have implications well beyond our newly proposed methods. By contrast, the results of Doikov et al. (2024) do not offer a new theoretical explanation for any already established method.

## 2 NOVEL STEPSIZE SCHEDULE

Now we are ready to present our new stepsize schedule based on the higher-order regularization.

**Theorem 1.** *For any $\sigma, \beta \geq 0$, the following adjustments of the Newton method are equivalent:*

$$\text{Regularization:} \qquad x^{k+1} = x^k + \operatorname*{argmin}_{y \in \mathbb{R}^d} T_{\sigma,\beta}\left(x^k\right), \tag{8}$$

$$\text{Damping:} \qquad x^{k+1} = x^k - \alpha_k [\nabla^2 f(x^k)]^{-1} \nabla f(x^k), \tag{9}$$

*where* $T_{\sigma,\beta}(x) = \operatorname{argmin}_{y \in \mathbb{R}^d} \left\{ \Phi_x(y) + \frac{\sigma}{2+\beta} \|y - x\|_x^{2+\beta} \right\}$ *and* $\alpha_k \in (0,1]$ *is the only positive root of polynomial* $P[\alpha] \stackrel{def}{=} 1 - \alpha - \alpha^{1+\beta}\sigma \|\nabla f(x^k)\|_{x^k}^{*\beta}$. *We call this algorithm Root Newton (RN), Algorithm 1.*

To simplify calculations, we reparametrize the RN via $\theta \stackrel{def}{=} \alpha^\beta \sigma \|\nabla f(x)\|_x^{*\beta}$, where $\theta \geq 0$ is an implicitly defined regularization constant. Using $\theta$, the polynomial $P$ simplifies to $P_\theta[\alpha] = 1 - \alpha - \alpha\theta$ and for any fixed $\theta$, the stepsize defined as $\alpha = \frac{1}{1+\theta}$ is the positive root of $P_\theta$. For a given iterate $x_k$ (and fixed $\beta$ and $\sigma$), $\theta$ and $\alpha$ are in one-to-one correspondence via $P_\theta$ (specifying either $\theta$ or $\alpha$ uniquely determines the other), so every admissible $\theta$ corresponds to a valid $\alpha$. Fortunately, we will be able to formulate our theory only in terms of $\theta$ (without $\alpha$), which will then in turn lead to guarantees for a corresponding stepsize $\alpha$.

## 2.1 HÖLDER CONTINUITY ASSUMPTION

Our analysis rely on assumption that the function has Hölder continuous Hessian or third derivative.

**Definition 1.** *For* $f : \mathbb{R}^d \to \mathbb{R}$, *and* $p \in \mathbb{N}$, *we say that p-times differentiable convex function is Hölder continuous of p-th order, if for some* $\nu \in [0,1]$ *there exists a constant* $L_{p,\nu} < \infty$, *so that*

$$\|\nabla^p f(x) - \nabla^p f(y)\|_{op} \leq L_{p,\nu} \|x - y\|_x^\nu, \qquad \forall x, y \in \mathbb{R}^d. \tag{10}$$

*We say that the f has Hölder continuous Hessian if* $L_{2,\nu} < \infty$ *(for some* $\nu \in [0,1]$*) and Hölder continuous third derivative if* $L_{3,\nu} < \infty$ *(for some* $\nu \in [0,1]$*).*

We would like to emphasize that Definition 1 is extremely general; the most general assumption for analysis of Newton methods. In particular, choice $L_{2,0}$ covers standard Lipschitz smoothness, $L_{3,0}$ covers constant bound on the third derivative, and $L_{2,1}$ is equivalent the semi-strong self-concordance (Hanzely et al., 2022). Further discussion of smoothness constants can be found in Section E. We will use the properties of the Hölder continuity summarized in the proposition below.

**Proposition 1.** *If function* $f : \mathbb{R}^d \to \mathbb{R}$ *is* $L_{2,\nu}$*-Hölder continuous, then it satisfies*

$$\left\| \nabla f(y) - \nabla f(x) - \nabla^2 f(x)[y - x] \right\|_x^* \leq \frac{L_{2,\nu}}{1+\nu} \|y - x\|_x^{1+\nu}.$$

*If function* $f : \mathbb{R}^d \to \mathbb{R}$ *is* $L_{3,\nu}$*-Hölder continuous, then it satisfies*

$$\left\| \nabla f(y) - \nabla f(x) - \nabla^2 f(x)[y - x] - \frac{1}{2}\nabla^3 f(x)[y - x]^2 \right\|_x^* \leq \frac{L_{3,\nu}}{(1+\nu)(2+\nu)} \|y - x\|_x^{2+\nu}.$$

To summarize the proof of the one-step loss decrease, we will show that Hölder continuity assumption with a sufficiently large regularization $\theta_k$ implies (for $c_1 \in \{1, 2\}$)

$$\left\langle \nabla f(x^{k+1}), \left[\nabla^2 f(x^k)\right]^{-1} \nabla f(x^k) \right\rangle \geq \frac{1}{2c_1(1-\alpha_k)} \|\nabla f(x^{k+1})\|_{x^k}^{*2},$$

which will in turn imply the one-step decrease as

$$f(x^k) - f(x^{k+1}) \geq -\left\langle \nabla f(x^{k+1}), x^{k+1} - x^k \right\rangle = \left\langle \nabla f(x^{k+1}), \alpha_k \left[\nabla^2 f(x^k)\right]^{-1} \nabla f(x^k) \right\rangle$$

$$\geq \frac{\alpha_k}{2c_1(1-\alpha_k)} \|\nabla f(x^{k+1})\|_{x^k}^{*2} = \frac{1}{2c_1\theta_k} \|\nabla f(x^{k+1})\|_{x^k}^{*2}. \tag{11}$$

Due to the level of technical detail, we defer lemmas for cases $p \in \{2, 3\}$ to Section A.2. We directly present their unification via reparametrization $q \stackrel{def}{=} p + \nu \in [2, 4]$, $M_q \stackrel{def}{=} L_{p,\nu}$.

**Theorem 2.** *Let $\|\nabla f(x)\|_x^* > 0$. Hölder continuity (Definition 1) with $p \in \{2,3\}, \nu \in [0,1]$ and $q = p + \nu$ for points $x^k, x^{k+1} = x^k - \alpha_k \left[\nabla^2 f(x^k)\right]^{-1} \nabla f(x^k)$, where $\alpha_k$ is the positive root of $P_{\theta_k}$. For $\theta_k$ such that*

$$\theta_k \geq (9M_q)^{\frac{1}{q-1}} \left\|\nabla f(x^k)\right\|_{x^k}^{* \frac{q-2}{q-1}} \tag{12}$$

*holds*

$$\left\langle \nabla f(x^{k+1}), \left[\nabla^2 f(x^k)\right]^{-1} \nabla f(x^k) \right\rangle \geq \frac{1}{2\alpha_k \theta_k} \left\|\nabla f(x^{k+1})\right\|_{x^k}^{*2}. \tag{13}$$

*In particular, in view of (11), we have that the choice $\theta_k = (9M_q)^{\frac{1}{q-1}} \left\|\nabla f(x^k)\right\|_{x^k}^{* \frac{q-2}{q-1}}$ guarantees decrease*

$$f(x^k) - f(x^{k+1}) \geq \frac{1}{2} \left(\frac{1}{9M_q}\right)^{\frac{1}{q-1}} \frac{\left\|\nabla f(x^{k+1})\right\|_{x^k}^{*2}}{\left\|\nabla f(x^k)\right\|_{x^k}^{* \frac{q-2}{q-1}}}. \tag{14}$$

Theorem 2 quantifies the amount of regularization $\theta_k$ needed for guaranteed decrease, leading to RN.

---

**Algorithm 1** RN: Root Newton stepsize schedule

---

1: **Requires:** Initial point $x^0 \in \mathbb{R}^d$, Hölder continuity exponent $q \in [2, 4]$ and constant $M_q < \infty$.
2: **for** $k = 0, 1, 2 \ldots$ **do**
3: $\quad n^k = \left[\nabla^2 f(x^k)\right]^{-1} \nabla f(x^k)$ $\qquad\qquad\qquad\qquad\qquad$ ▷ Newton direction
4: $\quad g_k = \left\langle \nabla f(x^k), n^k \right\rangle^{\frac{1}{2}}$ $\qquad\qquad\qquad\qquad\qquad$ ▷ $g_k = \left\|\nabla f(x^k)\right\|_{x^k}^*$
5: $\quad \theta_k = (9M_q)^{\frac{1}{q-1}} g_k^{\frac{q-2}{q-1}}$ $\qquad\qquad\qquad\qquad\qquad$ ▷ Sufficient regularization
6: $\quad \alpha_k = \frac{1}{1+\theta_k}$ $\qquad\qquad\qquad\qquad\qquad\qquad$ ▷ $\alpha_k$ is the root of $P_{\theta_k}[\alpha]$
7: $\quad x^{k+1} = x^k - \alpha_k n^k$ $\qquad\qquad\qquad\qquad\qquad$ ▷ Step, $x^k = T_{\sigma_k, \beta}(x^k)$
8: **end for**

---

## 2.2 Convergence garantees of RN

Global convergence guarantees are based on a standard assumption that the initial level set has finite diameter. Denote the initial level set $\mathcal{Q}(x^0) \stackrel{\text{def}}{=} \left\{x \in \mathbb{R}^d : f(x) \leq f(x^0)\right\}$ and its diameter as $D \stackrel{\text{def}}{=} \sup_{x,y \in \mathcal{Q}(x^0)} \|x - y\|_x$. Additionally, we need the Hessian not to change much between iterations.

**Assumption 1.** *For the sequence $\{x^k\}_{k=1}^\infty$, there exists a constant $\gamma > 0$ bounding Hessian of the consecutive points in gradient direction, $\gamma \leq \frac{\left\|\nabla f(x^{k+1})\right\|_{x^k}^{*2}}{\left\|\nabla f(x^{k+1})\right\|_{x^{k+1}}^{*2}}$.*

This assumption is also not novel, its variant has been used in Hanzely et al. (2022) for establishing local convergence as well as for analysis of quasi-Newton methods. Required $\gamma$ exists in many cases. For $L$-smooth $\mu$-strongly convex functions, $\gamma = \frac{\mu}{L}$. For functions with $\hat{c}$-stable Hessian (Karimireddy et al., 2018), $\gamma = \hat{c}$. For $L_{\text{sc}}$-self-concordant functions, it holds when iterates are close to each other (Nesterov & Nemirovski, 1994) or in the neighborhood of the solution (see proposition below).

**Proposition 2** (Hanzely et al. (2022), Lemma 4). *For convex $L_{sc}$-self-concordant function $f$ and iterate $x^k$ such that $\left\|\nabla f(x^k)\right\|_{x^k}^* \leq \frac{(2c_4+1)^2 - 1}{2L_{sc}}$ it holds $\nabla^2 f(x^{k+1})^{-1} \preceq (1 - c_4)^{-2} \nabla^2 f(x^k)^{-1}$.*

With assumptions clarified, we can jump straight to the convergence guarantees. First, we present superlinear local rate, which is expected for the stepsized Newton method.

**Theorem 3.** *Let function $f : \mathbb{R}^d \to \mathbb{R}$ be convex, Hölder continuous for $p \in \{2, 3\}, \nu \in [0, 1], q = p + \nu$ with $\gamma$-bounded Hessian change (1). Algorithm 1 has a superlinear local rate,*

$$\left\| \nabla f(x^{k+1}) \right\|_{x^{k+1}}^* \leq \frac{2}{\gamma} (9 M_q)^{\frac{1}{q-1}} \left\| \nabla f(x^k) \right\|_{x^k}^{*\left(2 - \frac{1}{q-1}\right)}.$$

For the $L_{2,1}$-Hölder continuous functions, the presented rate is suboptimal compared to quadratic rate of AICN schedule (4). However, the rate of Theorem 3 holds for any $q$, and its exponent increases with $q$ (up to $5/3$ for $q = 4$).

For global convergence guarantees, we first quantify in general the decrease implied by Theorem 2. This will provide plug-in guarantees for the RN and other algorithms.

**Lemma 2.** *Let function $f : \mathbb{R}^d \to \mathbb{R}$ be convex with $\gamma$-bounded Hessian change (1) and the bound level sets with diameter $D$. If an algorithm $\mathcal{A}$ generates the iterates $\left\{ x^k \right\}_{k=1}^K$ with one-step decrease for $q \geq 2$ and $c_5 \geq 0$ as*

$$f(x^k) - f(x^{k+1}) \geq c_5 \frac{\left\| \nabla f(x^{k+1}) \right\|_{x^k}^{*2}}{\left\| \nabla f(x^k) \right\|_{x^k}^{*\frac{q-2}{q-1}}}, \tag{15}$$

*then $\mathcal{A}$ has the global convergence rate $f(x^K) - f_* \leq D \left( \frac{2(q-1)D}{\gamma c_5 K} \right)^{q-1} + \left\| \nabla f(x^0) \right\|_{x^0}^* D e^{-K/4}$.*

**Theorem 4.** *Let function $f : \mathbb{R}^d \to \mathbb{R}$ be convex, Hölder continuous for $p \in \{2, 3\}, \nu \in [0, 1], q = p + \nu$ with $\gamma$-bounded Hessian change (1) and the bound level sets with diameter $D < \infty$. Algorithm 1 (RN) with known parameters $q, M_q$ converges with rate $\mathcal{O}\left( \frac{M_q D^q}{k^{q-1}} \right)$ as*

$$f(x^k) - f_* \leq 9 M_q D \left( \frac{4D(q-1)}{\gamma k} \right)^{q-1} + \left\| \nabla f(x^0) \right\|_{x^0}^* D e^{-k/4}.$$

Algorithm RN also achieves global linear and superlinear convergence rates under different assumptions. Due to the space constraints, we deferred these results to Section B.

Note that the loss function can satisfy Hölder continuity (Definition 1) with multiple different $L_{p,\nu}$, and therefore different pairs $(q, M_q)$ can be used. The best parametrization might not be known.

## 3 UNKNOWN PARAMETRIZATION

To address unknown parameterization, we propose to not rely on Theorem 2, but to find a point maximizing the bound (15) directly,

$$x^{k+1} = \underset{y \in \left\{ x - \alpha n_{x^k} | \alpha \in [0,1] \right\}}{\operatorname{argmin}} \frac{f(y) - f(x^k)}{\left\| \nabla f(y) \right\|_{x^k}^{*2}}, \tag{16}$$

where $n_x \stackrel{\text{def}}{=} [\nabla^2 f(x)]^{-1} \nabla f(x)$ is a shorthand for Newton's direction at point $x$. We call this algorithm Gradient-Regulated Line Search (GRLS, Algorithm 4). Interestingly, this theoretical linesearch simultaneously minimizes loss and gradient norms. Its rate follows directly from Lemma 2.

**Corollary 1.** *Let function $f : \mathbb{R}^d \to \mathbb{R}$, be convex, Hölder continuous with some $M_q < \infty$, with $\gamma$-bounded Hessian change (1), and the bound level sets with diameter $D < \infty$. Linesearch GRLS converges as $f(x^k) - f_* \leq \min_{q \in [2,4]} 9 M_q D \left( \frac{4D(q-1)}{\gamma k} \right)^{q-1} + \left\| \nabla f(x^0) \right\|_{x^0}^* D e^{-k/4}$.*

Observe that for small stepsizes $\alpha_k \in [0, \overline{\alpha}]$, for some $\overline{\alpha} \ll 1$, model differences are small $x^{k+1} \approx x^k$ and $\nabla f(x^k) \approx \nabla f(x^{k+1})$. Therefore, expression (16) minimized by GRLS can be approximated as

$$\frac{f(y) - f(x^k)}{\left\| \nabla f(y) \right\|_{x^k}^{*2}} \approx \frac{f(y) - f(x^k)}{\left\| \nabla f(x^k) \right\|_{x^k}^{*2}}, \tag{17}$$

and the right-hand-side is minimized by the popular Newton method with greedy linesearch,

$$x^{k+1} = \underset{y \in \{x^k - \alpha n_{x^k} | \alpha \in [0,1]\}}{\operatorname{argmin}} f(y), \tag{18}$$

which we will call *Greedy Newton* (GN). Our experimental evaluations will demonstrate that linesearches GN and GRLS use similar stepsizes (Figures 2c, 3c) justifying (17). Therefore while GRLS enjoys strong convergence guarantees, method GN is preferable in practice due to its easier criterion. Nevertheless, this connection allows us to obtain the convergence rate for the Greedy Newton in the corollary below. We refer the reader for more detailed explanation to Section C.

**Corollary 2.** *Let function* $f : \mathbb{R}^d \to \mathbb{R}$, *be convex,* $M_q$-*Hölder continuous for some* $M_q < \infty$, *with* $\gamma$-*bounded Hessian change* (1), *and the bound level sets with diameter* $D < \infty$. *If the Greedy Newton linesearch* (18) *satisfies the inequality* $\left\|\nabla f(x^{k+1})\right\|_{x^k}^* \leq \bar{c}\left\|\nabla f(x^k)\right\|_{x^k}^*$ *with some constant* $\bar{c} \geq 0$ *for all iterates* $x^k$, *then it has convergence guarantee* $\min_{q \in [2,4]} \mathcal{O}\left(\frac{M_q D^q \bar{c}^{2(q-1)}}{k^{q-1}}\right)$

$$f(x^k) - f_* \leq \min_{q \in [2,4]} 9 M_q D \left(\frac{4 D \bar{c}^2 (q-1)}{\gamma k}\right)^{q-1} + \left\|\nabla f(x^0)\right\|_{x^0}^* D e^{-k/4}.$$

**Remark.** *Corollary 2 introduces assumption that the gradients norm measured in the local norms does not increase by more than a constant factor in between the iterates,* $\left\|\nabla f(x^{k+1})\right\|_{x^k}^* \leq \bar{c}\left\|\nabla f(x^k)\right\|_{x^k}^*$. *For any sequence* $\{x_k\}_{k=1}^\infty$ *monotonically decreasing loss* $f$, *this holds for example for quadratic functions with constant* $\bar{c}$.

In this section, we established fast convergence guarantees for the novel but impractical linesearch method GRLS (16) and for the popular GN scheme (18), both of which do not require prior knowledge of the smoothness parameters $(q, M_q)$. However, their implicit nature may not be suitable for all practical scenarios. To address this limitation, in the next section we introduce a practical stepsize backtracking procedure with matching convergence guarantees.

## 4 UNIVERSAL STEPSIZE BACKTRACKING

Our backtracking procedure is based on the observation that the knowledge of the parametrization $(q, M_q)$ in RN is required only for setting $\theta_k$. We start with an estimate of $\theta_k$ smaller than the true value and increase it until it achieves the theoretically predicted decrease. We claim that the resulting algorithm UN is well-defined with a bounded number of backtracking steps.

To formalize this claim, we quantify the smallest plausible true $\theta_k$ that will be estimated first. For $q \in [2,4]$ and $\beta \geq \frac{2}{3}$ denote $\mathcal{H}(x) \stackrel{\text{def}}{=} \inf_{q \in [2,4]} (9 M_q)^{\frac{1}{q-1}} \|\nabla f(x)\|_x^{*\left(\frac{q-2}{q-1} - \beta\right)}$.

**Lemma 3.** *If* $M_q < \infty$ *for some* $q \in [2,4]$, *and the initial estimate* $\sigma_0$ *small enough,* $\sigma_0 \leq \mathcal{H}(x^0)$, *then all iterations* $\{x^k\}_{k=0}^n$ *of* UN, *such that* $\left\|\nabla f(x^k)\right\|_{x^k}^* > 0$, *satisfy* $\sigma_{k+1} = \frac{\theta_{k,j_k-1}}{\|\nabla f(x^k)\|_{x^k}^{*\beta}} \leq \mathcal{H}(x^k)$. *Moreover, the total number of backtracking steps during the first* $k$ *iterations,* $N_K$, *is bounded as* $N_k \leq 2k + \log_\rho\left(\mathcal{H}(x^{k-1})/\sigma_0\right)$.

**Theorem 5.** *Let function* $f : \mathbb{R}^d \to \mathbb{R}$, *be convex, Hölder continuous for* $p \in \{2,3\}, \nu \in [0,1], q = p + \nu$ *with bounded Hessian change (Assumption 1) and the bound level sets diameter* $D < \infty$. *Algorithm 2 (*UN*) converges with the rate* $\min_{q \in [2,4]} \mathcal{O}\left(\frac{M_q D^q}{k^{q-1}}\right)$,

$$f(x^k) - f_* \leq \min_{q \in [2,4]} 9 M_q D \left(\frac{4 \rho^2 D (q-1)}{k}\right)^{q-1} + \left\|\nabla f(x^0)\right\|_{x^0}^* D e^{-k/4}.$$

## 5 RESULTS OF NUMERICAL EXPERIMENTS

The majority of figures and the detailed technical description were deferred to Section A.1.

---

**Algorithm 2** UN: Universal stepsize backtracking procedure for the Newton method

---

1: **Input:** Initial point $x^0 \in \mathbb{R}^d$, constants $\sigma_0 > 0, \rho > 1, \rho \geq \gamma^{-\frac{2}{3}}, \beta \in \left[\frac{2}{3}, 1\right]$

$\qquad\qquad\qquad\qquad\qquad\qquad\qquad \triangleright$ Note $\beta \geq \frac{q-2}{q-1}, \rho \geq \gamma^{-\frac{q-2}{q-1}}$ for $q \in [2, 4]$

2: **for** $k = 0, 1, 2 \ldots$ **do**

3: $\qquad n^k = \left[\nabla^2 f(x^k)\right]^{-1} \nabla f(x^k)$ $\qquad\qquad\qquad\qquad\qquad\qquad \triangleright$ Newton direction

4: $\qquad g_k = \left\langle \nabla f(x^k), n^k \right\rangle^{\frac{1}{2}}$ $\qquad\qquad\qquad\qquad\qquad\qquad\qquad \triangleright g_k = \left\| \nabla f(x^k) \right\|_{x^k}^*$

5: $\qquad$ **for** $j_k = 0, 1, 2 \ldots$ **do**

6: $\qquad\qquad \theta_{k,j_k} = \rho^{j_k} \sigma_k g_k^{\beta}$ $\qquad\qquad\qquad\qquad\qquad\qquad\qquad \triangleright$ Increase regularization

7: $\qquad\qquad \alpha_{k,j_k} = \frac{1}{1+\theta_{k,j_k}}$ $\qquad\qquad\qquad\qquad\qquad\qquad\qquad\qquad \triangleright$ Update stepsize

8: $\qquad\qquad x_{j_k}^k = x^k - \alpha_{k,j_k} n^k$ $\qquad\qquad\qquad\qquad\quad \triangleright$ Step, $x_{j_k}^k = T_{\rho^{j_k}\sigma_k, \beta_k}\left(x^k\right)$

9: $\qquad\qquad$ **if** $\left\langle \nabla f(x_{j_k}^k), n^k \right\rangle \geq \frac{1}{2\alpha_{k,j_k}\theta_{k,j_k}} \left\| \nabla f(x_{j_k}^k) \right\|_{x^k}^{*2}$ **then**

10: $\qquad\qquad\qquad x^{k+1} = x_{j_k}^k$

11: $\qquad\qquad\qquad \sigma_{k+1} = \rho^{j_k-1} \sigma_k$

12: $\qquad\qquad\qquad$ **break**

13: $\qquad\qquad$ **end if**

14: $\qquad$ **end for**

15: **end for**

---

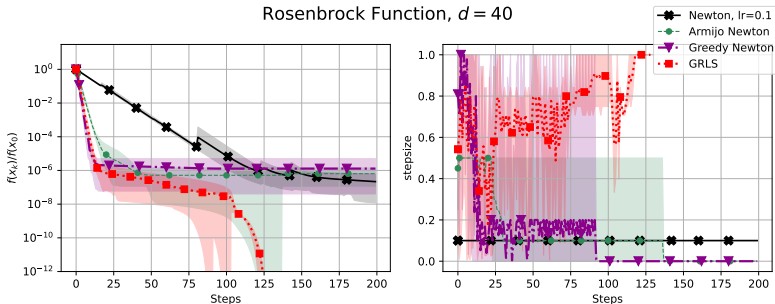

Figure 1: Performance of Newton method stepsize linesearch procedures on the notoriously challenging nonconvex **Rosenbrock function** (21). We plot mean $\pm$ standard deviation of 5 random initializations. We crop stepsize standard deviation at 0.

In Figures 2a, 3a, we compare higher-order methods *without* any linesearch procedures, namely RN, AICN (Hanzely et al., 2022) and Gradient Regularization of Newton Method (GRN) (Doikov et al., 2024, Alg. 1). As additional baselines, we use the damped Newton method with a fixed fine-tuned stepsize and classical first-order Gradient Method (GM) (Nesterov, 2018). RN and AICN show similar performance while GRN has a slight disadvantage. Unsurprisingly, the first-order method GM has quicker iterations but slower per-iteration convergence.

In Figures 2b, 3b, we compare higher-order regularization methods *with* smoothness constant estimation procedures, UN and Super-universal Newton method (Doikov et al., 2024, Alg. 2). As an additional baseline, we use the damped Newton method with a fixed but fine-tuned stepsize. We show that UN displays faster convergence than the Super-universal Newton method. Moreover, we show that the exponent of the regularization term $\beta$ that appears in both UN and super-universal Newton method (6) does not have a significant impact on overall performance.

Figures 2c, 3c, 1 compare implicit linesearch procedures for Newton stepsizes, namely GRLS, Armijo stepsize, and Greedy Newton stepsize (GN) (Cauchy, 1847; Shea & Schmidt, 2025). Our theory presents convergence guarantees for GRLS and GN with stepsizes limited to the interval $[0, 1]$. We go beyond this limitation and perform parameter linesearches over $\alpha \in \mathbb{R}_+$ instead.

Figures 2c, 3c demonstrate that on logistic regression and polytope feasibility problems, linesearch procedures GRLS and GN use almost indistinguishable stepsizes and converge faster than Armijo linesearch and fixed stepsize Newton. On the Rosenbrock function (Figure 1), GRLS outperforms all other linesearches procedures.

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

## A  DETAILED DESCRIPTIONS

### A.1  DETAILED DESCRIPTIONS OF EXPERIMENTS

#### LOGISTIC REGRESSION LOSS

In Figure 2, we compare the performance of the proposed algorithms on binary classification on datasets from LIBSVM repository (Chang & Lin, 2011). For datapoints $\{(a_i, b_i)\}_{i=1}^n$, where $a_i \in \mathbb{R}^d, b_i \in \{-1, +1\}$, and regularizer $\mu = 10^{-3}$, we aim to minimize

$$\min_{x \in \mathbb{R}^d} \left\{ f(x) = \frac{1}{n} \sum_{i=1}^n \log\left(1 + e^{-b_i \langle a_i, x \rangle}\right) + \frac{\mu}{2} \|x\|_2^2 \right\}. \tag{19}$$

We initialize all methods at $x_0 = 10 \cdot [1, 1, \ldots, 1]^T \in \mathbb{R}^d$.

#### POLYTOPE FEASIBILITY LOSS

In Figure 3, we compare proposed algorithms on *polytope feasibility* problem, aiming to find a point from a polytope $\mathcal{P} = \left\{ x \in \mathbb{R}^d : \langle a_i, x \rangle \le b_i, \ 1 \le i \le n \right\}$, reformulated as

$$\min_{x \in \mathbb{R}^d} \left\{ f(x) = \sum_{i=0}^n (\langle a_i, x \rangle - b_i)_+^p \right\}, \tag{20}$$

where $(t)_+ \overset{\text{def}}{=} \max\{t, 0\}$ and $p \ge 2$. We generate data points $(a_i, b_i)$ and the solution $x^*$ synthetically as $a_i, x^* \sim \mathcal{N}(0, 1)$ and set $b_i = \langle a_i, x^* \rangle$.

We initialize all methods at $x_0 = [1, 1, \ldots, 1]^T \in \mathbb{R}^d$.

#### ROSENBROCK LOSS

Linesearch procedures solve the abovementioned problems in just a few steps. For a more challenging task, Figure 1 presents the notorious $d$-dimensional *Rosenbrock* function,

$$\min_{x \in \mathbb{R}^d} \left\{ f(x) = \sum_{i=0}^{d-1} [100(x_{i+1} - x_i^2)^2 + (1 - x_i)^2] \right\}. \tag{21}$$

Notably, the Rosenbrock function (21) is nonconvex, which breaks assumptions in our convergence theorems.

The function (21) has the global solution at $x^* = [1, \ldots, 1]^T$, and therefore we choose the initial point from a normal distribution, $x^0 \sim \mathcal{N}(0, I_d) \cdot 20$.

#### TECHNICAL DETAILS

All hyperparameters were fine-tuned to achieve the best possible performance for both objectives and every dataset. All experiments were conducted on a workstation with specifications: AMD EPYC 7742 64-Core Processor with 32Gb of RAM. Source code is available at https://github.com/fxrshed/root-newton.

#### A.1.1  EXTENDED COMPARISON ON ROSENBROCK FUNCTION

In Figure 4 we present an extended comparison of linesearch procedures on Rosenbrock function (21) (similar to Figure 1), with 10 random initializations and the limit of 1000 steps. We observe that none of the considered algorithms consistently converge to the exact solution for all of the random seeds, and that GRLS performs better than the other linesearch methods.

### A.2  HÖLDER CONTINUITY TO ONE STEP DECREASE

**Lemma 4.** *Let* $\left\|\nabla f(x^k)\right\|_{x^k}^* > 0$, *and* $x^k \in \mathbb{R}^d$, $x^{k+1} = x^k - \alpha_k \left[\nabla^2 f(x^k)\right]^{-1} \nabla f(x^k)$, *as in* RN.

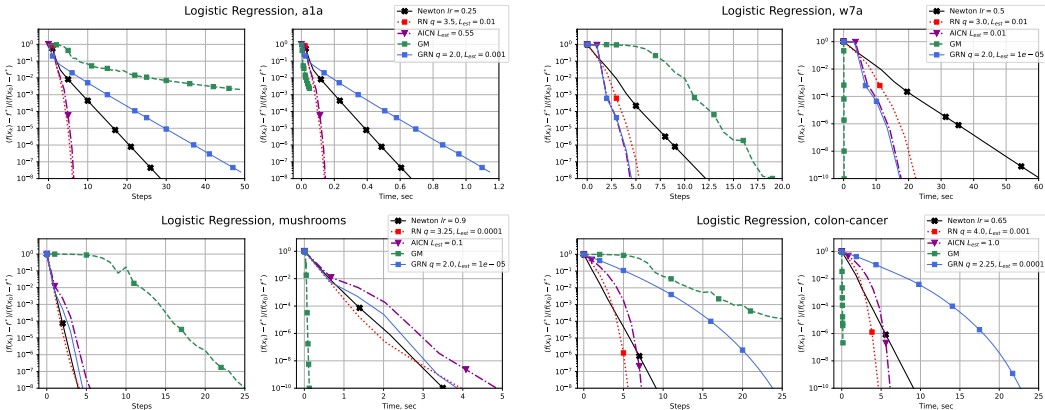

(a) Performance of RN compared to other higher-order methods *without* any linesearch procedure.

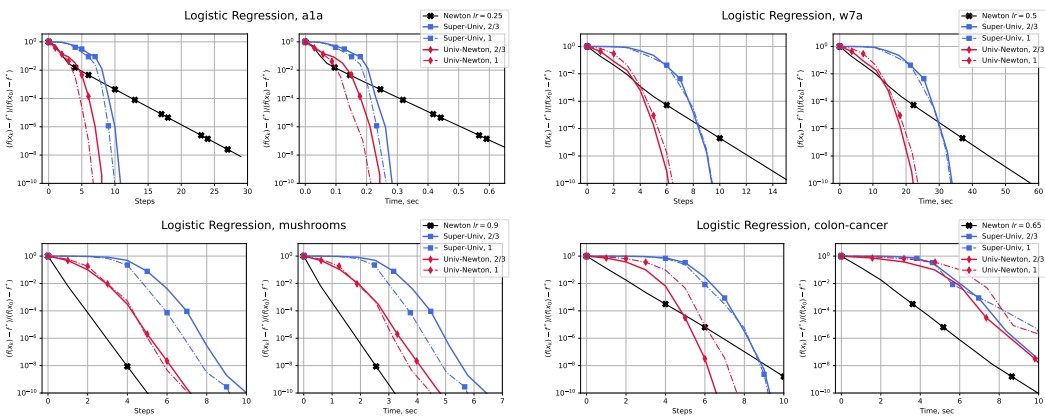

(b) Performance of UN compared to other higher-order regularization methods *with* smoothness estimation procedures.

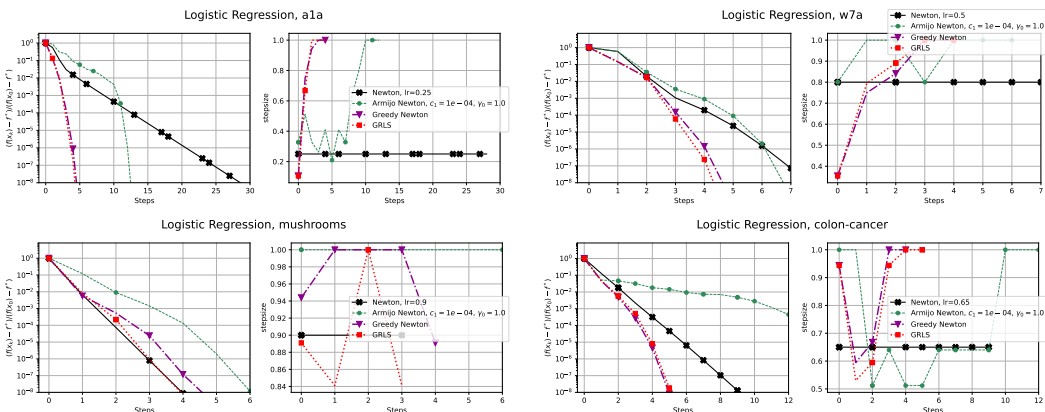

(c) Performance of Linesearch GRLS (16) compared to other linesearch procedures.

Figure 2: Binary classification **logistic regression** problem on LIBSVM datasets.

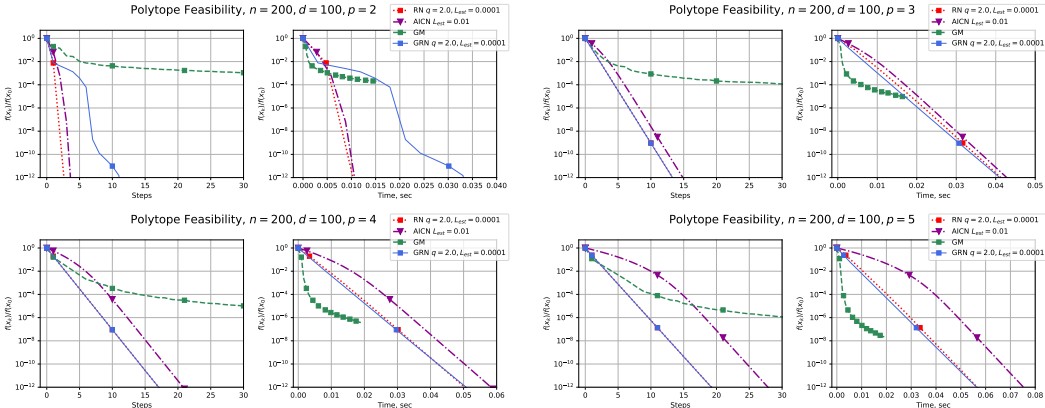

(a) Performance of RN compared to other higher-order methods *without* any linesearch procedure.

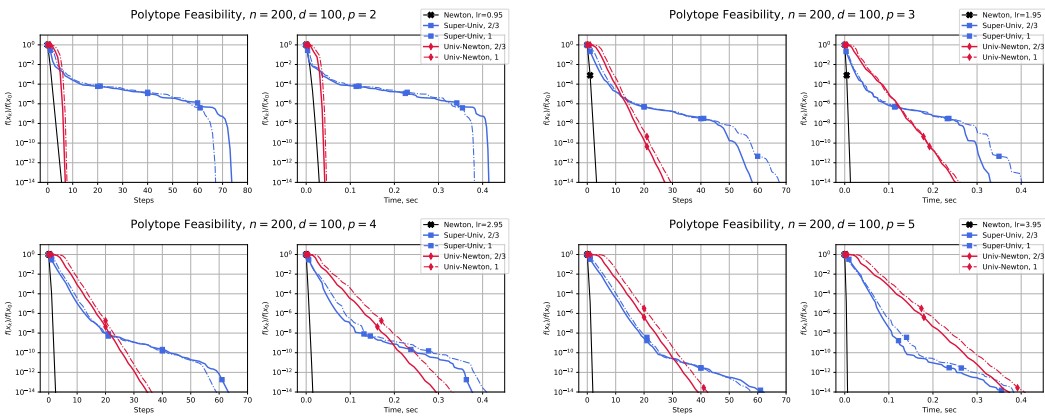

(b) Performance of UN compared to other higher-order regularization methods *with* smoothness estimation procedures.

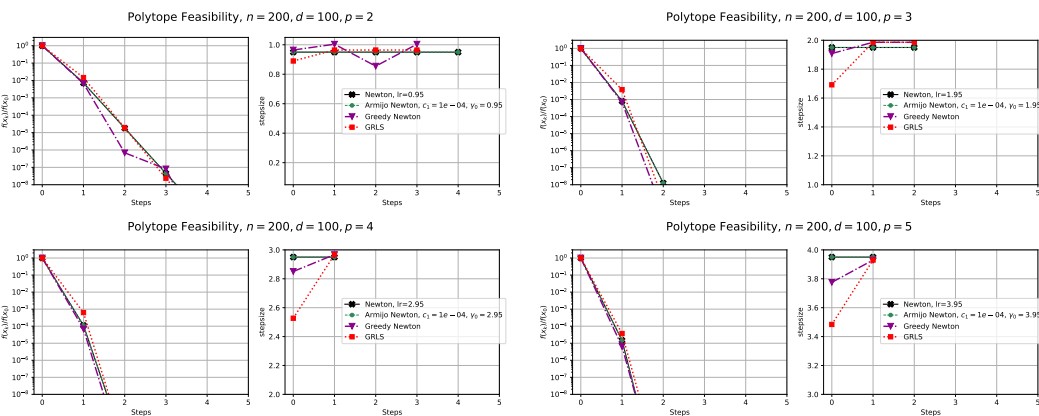

(c) Performance of Linesearch GRLS (16) compared to other linesearch procedures.

Figure 3: **Polytope feasibility** problem (20) on a synthetic datasets.

---

**Algorithm 3** Line search backtracking procedures for the Newton direction

---

1: **Inputs:** Initial learning rate $\gamma_0 > 0$, constants $c_1, c_2 \in (0, 1)$, shrinkage factor $\rho \in (0, 1)$, current iterate $x \in \mathbb{R}^d$, termination condition $C$ defined as

$$C(x_+, x) \leftarrow$$
$$\begin{cases} f(x_+) \leq f(x) - c_1\gamma\|\nabla f(x)\|_x^{*2} & \text{Armijo} \\ f(x_+) \leq f(x) - c_1\gamma\|\nabla f(x)\|_x^{*2} \ \& \ \langle n, \nabla f(x_+)\rangle \leq c_2\|\nabla f(x)\|_x^{*2} & \text{Wolfe} \\ f(x_+) \leq f(x) - c_1\gamma\|\nabla f(x)\|_x^{*2} \ \& \ |\langle n, \nabla f(x_+)\rangle| \leq c_2\|\nabla f(x)\|_x^{*2} & \text{Strong Wolfe} \end{cases}$$

2: Compute Newton's direction $n_x \leftarrow -\left[\nabla^2 f(x)\right]^{-1}\nabla f(x)$
3: Initialize $\gamma \leftarrow \gamma_0$
4: **while** $C(x + \gamma n_x, x)$ is not satisfied **do**
5: $\quad \gamma \leftarrow \rho\gamma$
6: **end while**
7: Return next point $x + \gamma n_x$

---

**Algorithm 4** GRLS: Gradient Regularized Line Search

---

1: **Requires:** Initial point $x^0 \in \mathbb{R}^d$.
2: **for** $k = 0, 1, 2 \ldots$ **do**
3: $\quad n^k = \left[\nabla^2 f(x^k)\right]^{-1}\nabla f(x^k)$ $\quad\quad\quad\quad\quad\quad\quad\quad\quad$ ▷ Newton direction
4: $\quad$ Compute next iterate

$$x^{k+1} = \operatorname*{argmin}_{y \in \{x - \alpha n^k | \alpha \in [0,1]\}} \frac{f(y) - f(x^k)}{\|\nabla f(y)\|_{x^k}^{*2}}$$

5: **end for**

---

● *Hölder continuity of **Hessian** (Def. 1 with $p = 2$) implies that for $\theta_k$ larger than $\theta_k \geq \frac{L_{2,\nu}}{1+\nu}\alpha_k^\nu\|\nabla f(x^k)\|_{x^k}^{*\nu}$ holds*

$$\left\langle \nabla f(x^{k+1}), \left[\nabla^2 f(x^k)\right]^{-1}\nabla f(x^k)\right\rangle \geq \frac{1}{2(1-\alpha_k)}\|\nabla f(x^{k+1})\|_{x^k}^{*2}.$$

● *Hölder continuity of the **third derivative** (Definition 1 with $p = 3$) implies that for regularization $\theta_k$ larger than*

$$\theta_k \geq \alpha_k\|\nabla f(x^k)\|_{x^k}^* \max\left\{6\left(\frac{L_{3,\nu}}{1+\nu}\right)^{\frac{1}{1+\nu}}, \ \frac{\sqrt{3}L_{3,\nu}}{(1+\nu)(2+\nu)}\left(\alpha_k\|\nabla f(x^k)\|_{x^k}^*\right)^\nu\right\}, \quad (22)$$

*holds*

$$\left\langle \nabla f(x^{k+1}), \left[\nabla^2 f(x^k)\right]^{-1}\nabla f(x^k)\right\rangle \geq \frac{1}{4(1-\alpha_k)}\|\nabla f(x^{k+1})\|_{x^k}^{*2}.$$

In Lemma 4, requirements on $\theta_k$ are inconveniently dependent on $\alpha_k$. We can use the following observation to derive a bound dependent only on the norm of the gradient.

**Lemma 5.** *For $c_3, \delta > 0$, choice $\theta_k \geq c_3^{\frac{1}{1+\delta}}\|\nabla f(x^k)\|_{x^k}^{*\frac{\delta}{1+\delta}}$ ensures $\theta_k \geq c_3\left(\alpha_k\|\nabla f(x^k)\|_{x^k}^*\right)^\delta$.*

## B  GLOBAL (SUPER)LINEAR CONVERGENCE RATE

Stepsized Newton method is known to be able to achieve a global linear rate if the Hessian is bounded and stepsize is constant (Karimireddy et al., 2018; Gower et al., 2019b), or when the function is $L_{2,1}$-Hölder continuous with stepsize following schedule AICN (Hanzely et al., 2022, proof in (Hanzely, 2025)).

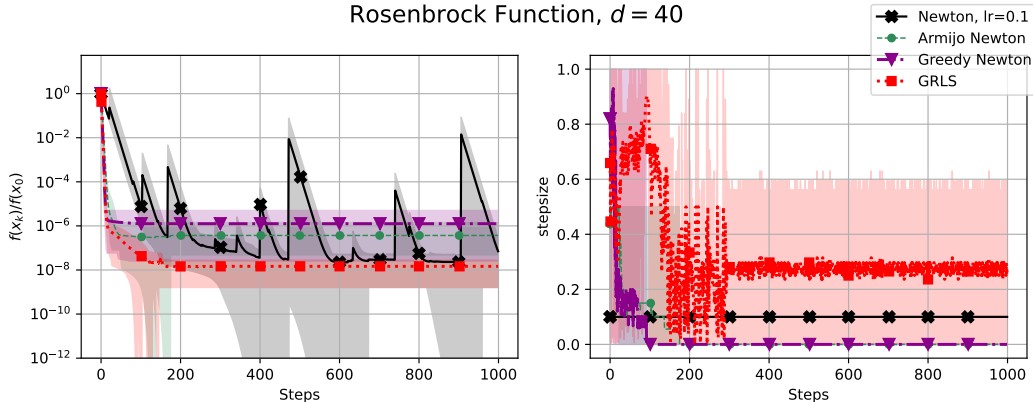

Figure 4: Performance of Newton method stepsize linesearch procedures on nonconvex **Rosenbrock function** (21). We plot mean $\pm$ standard deviation of 10 random initializations. We crop stepsize standard deviation at 0.

In line with those results, we present global linear rates for algorithms RN, UN, GRLS on $L_{p,\nu}$-Hölder continuous functions with finite $(p + \nu)$-relative size characteristic (Doikov et al., 2024). The proof is in Section G.

**Definition 2** ((Doikov et al., 2024)). *For strictly convex function $f : \mathbb{R}^d \to \mathbb{R}$ we call $s$-relative size characteristic*

$$D_s \stackrel{def}{=} \sup_{x,y \in \mathcal{Q}(x^0)} \left\{ \|x - y\|_x \left( \frac{V_f}{\beta_f(x,y)} \right)^{\frac{1}{s}} \right\},$$

*where $\beta_f(x,y) \stackrel{def}{=} \langle \nabla f(x) - \nabla f(y), x - y \rangle > 0$ and $V_f \stackrel{def}{=} \sup_{x,y \in \mathcal{Q}(x^0)} \beta_f(x,y)$.*

**Theorem 6.** *Let function $f$ be $L_{p,\nu}$-Hölder continuous, with finite relative size $D_q < \infty$ for $q = p + \nu$ (Definition 2) and $\gamma$-bounded Hessian change (Assumption 1). Algorithms RN, UN and GRLS find points in the $\varepsilon$-neighborhood, $f(x^k) - f(x^*) \leq \varepsilon$, in*

$$k \leq \mathcal{O}\left( \gamma \left( \frac{M_q D_q^q}{V_f} \right)^{\frac{1}{q-1}} \ln \frac{f_0}{\varepsilon} + \ln \frac{\|\nabla f(x^0)\|_{x^0}^* D}{\varepsilon} \right)$$

*iterations, implying a global linear convergence rate.*

**Remark.** *In view of (17), analogous convergence guarantee (with a worse constant) can be proven for GN.*

Replacing relative size assumption with uniform star-convexity of degree $s$ ($q > s \geq 2$), we can guarantee a global superlinear rate for RN and GN similarly to Kamzolov et al. (2025).

**Definition 3.** *For $s \geq 2$ and $\mu_s \geq 0$ we call function $f : \mathbb{R}^d \to \mathbb{R}$ $\mu_s$-uniformly star-convex of degree $s$ in local norms with respect to a minimizer $x^*$ if $\forall x \in \mathbb{R}^d, \forall \eta \in [0, 1]$ holds*

$$f\left( \eta x + (1 - \eta)x^* \right) \leq \eta f(x) + (1 - \eta)f_* - \frac{\eta(1 - \eta)\mu_s}{s} \|x - x^*\|_x^s.$$

*If this inequality holds for $\mu_s = 0$, we call function $f$ star-convex in local norms (w.r.t. minimizer $x^*$).*

**Theorem 7.** *Let the function $f : \mathbb{R}^d \to R$ be $L_{p,\nu}$-Hölder continuous (Definition 1) and $\mu_s$-uniformly star-convex of degree $s$ in local norms (Definition 3) and $q \stackrel{def}{=} p + \nu \geq s \geq 2$ then*

RN *and* GN *have global decrease in functional value suboptimality,*

$$f(x^k) - f_* \leq \left(f(x^0) - f_*\right) \prod_{t=0}^{k-1}(1 - \hat{\eta}_t),$$

*where* $\hat{\eta}_k \in [0,1]$ *is the only positive root of* $E_k(\eta) \overset{\text{def}}{=} (1 - \eta)\frac{\mu_s}{s} - \eta^{q-1}\left(\frac{M_q}{(p+1)!} + \frac{\sigma}{q}\right)\left\|x^k - x^*\right\|_{x^k}^{q-s}.$

*If $q = s$, then $\hat{\eta}_k$ is constant throughout all iterations and the rate is **globally linear**.*

*If $q > s$, then $\hat{\eta}_k$ is monotonically increasing as $\left\|x^k - x^*\right\|_{x^k}$ decreases, $1 - \hat{\eta}_k \to 0$, and therefore, the resutling rate is **globally superlinear**.*

*Proof of Theorem 7.* We have that updates of RN with $q = p + \nu = 2 + \beta$ and any $\sigma \geq M_q$ can be written as

$$f(x^{k+1}) \leq \Phi_{x^k}(x^{k+1}) + \frac{\sigma}{q}\left\|x^{k+1} - x^k\right\|_{x^k}^q \tag{23}$$

$$= \min_{y \in \mathbb{R}^d}\left\{\Phi_{x^k}(y) + \frac{\sigma}{q}\|y - x\|_{x^k}^q\right\}, \tag{24}$$

using standard integration arguments from $M_q$-Hölder continuity

$$\leq \min_{y \in \mathbb{R}^d}\left\{f(y) + \frac{M_q}{(p+1)!}\|y - x^k\|_{x^k}^q + \frac{\sigma}{q}\|y - x^k\|_{x^k}^q\right\} \tag{25}$$

$$= \min_{y \in \mathbb{R}^d}\left\{f(y) + \left(\frac{M_q}{(p+1)!} + \frac{\sigma}{q}\right)\|y - x^k\|_{x^k}^q\right\}, \tag{26}$$

setting $y \leftarrow x + \eta_k(x^* - x^k)$ for arbitrary $\eta_k \in [0,1]$,

$$\leq f\left(x^k + \eta_k(x^* - x^k)\right) + \eta_k^q\left(\frac{M_q}{(p+1)!} + \frac{\sigma}{q}\right)\left\|x^k - x^*\right\|_{x^k}^q, \tag{27}$$

assuming $\mu_s$-strong star-convexity for $q \geq s \geq 2$,

$$\leq (1 - \eta_k)f(x^k)$$
$$+ \eta_k f_* - \frac{\eta_k(1 - \eta_k)\mu_s}{s}\left\|x^k - x^*\right\|_{x^k}^s + \eta_k^q\left(\frac{M_q}{(p+1)!} + \frac{\sigma}{q}\right)\left\|x^k - x^*\right\|_{x^k}^q, \tag{28}$$

denoting functional suboptimality $\delta_k \overset{\text{def}}{=} f(x^k) - f_*$,

$$\delta_{k+1} \leq (1 - \eta_k)\delta_k$$
$$- \eta_k\left\|x^k - x^*\right\|_{x^k}^s\left((1 - \eta_k)\frac{\mu_s}{s} - \eta_k^{q-1}\left(\frac{M_q}{(p+1)!} + \frac{\sigma}{q}\right)\left\|x^k - x^*\right\|_{x^k}^{q-s}\right). \tag{29}$$

Denote expression $E(\eta) \overset{\text{def}}{=} (1 - \eta)\frac{\mu_s}{s} - \eta^{q-1}\left(\frac{M_q}{(p+1)!} + \frac{\sigma}{q}\right)\|x - x^*\|_x^{q-s}$ for $\eta \in [0,1]$. Observe that $E'(\eta) < 0$ and therefore $E$ is monotonically decreasing on $\mathbb{R}^+$; with $E(0) \geq 0 \leq E(1)$ we can conclude that it has a unique root $\hat{\eta}$ on $[0,1]$. With choice $\eta \leftarrow \hat{\eta}$ in the last inequality we can conclude global convergence rate

$$\delta_{k+1} \leq (1 - \hat{\eta}_k)\delta_k. \tag{30}$$

Note that the root of the expression $E$ is inversely proportional to the distance from the solution $\|x - x^*\|_x$, and therefore as the method converges, $x^k \to x^*$, then the size of its root increases $\hat{\eta}_k \to 1$. Therefore, the global convergence rate (30) is superlinear.

Unrolling the recurrence (30) yields the inequality from the Theorem 7.

Note that the decrease is based solely on the decrease in functional values, which allows us to prove the identical guarantee for Greedy Newton linesearch GN. In particular, GN implies $f(x_{\mathsf{GN}}^+) \leq f(x_{\mathsf{RN}}^+)$, and we can analogically conclude

$$f(x_{\mathsf{GN}}^{k+1}) - f_* \leq \left(f(x_{\mathsf{GN}}^k) - f_*\right)(1 - \hat{\eta}_k). \tag{31}$$

$\square$

## C  FAST CONVERGENCE GUARANTEES FOR GREEDY NEWTON LINESEARCH

If the inequality $\|\nabla f(y)\|_{x^k}^* \leq \bar{c}\|\nabla f(x^k)\|_{x^k}^*$ holds for constant $\bar{c} \geq 0$, we have that for stepsizes in a range $[\underline{\alpha}, \overline{\alpha}]$ holds

$$\min_{\substack{\alpha \in [\underline{\alpha}, \overline{\alpha}] \\ y = x - \alpha n_{x^k}}} \frac{f(y) - f(x^k)}{\|\nabla f(x^k)\|_{x^k}^{*2}} \leq \bar{c}^2 \cdot \min_{\substack{\alpha \in [\underline{\alpha}, \overline{\alpha}] \\ y = x - \alpha n_{x^k}}} \frac{f(y) - f(x^k)}{\|\nabla f(y)\|_{x^k}^{*2}}, \tag{32}$$

proving that Greedy Newton minimizes the target metric of GRLS up to a constant $\times \bar{c}^2$. If we denote $\hat{c}_5$ constant with which GRLS satisfies Lemma 2, then Greedy Newton satisfies Lemma 2 with constant $\hat{c}_5\bar{c}^2$ and guarantee convergence similar to Corollary 1.

Now we are going to discuss how constant $\bar{c}$ can be found in different scenarios.

**Remark** (General $M_q$-Hölder continuous functions). *To find $\bar{c}$ we note that Theorem 2 shows that stepsize $\theta_k \stackrel{def}{=} \frac{1-\alpha_k}{\alpha_k} \geq (9M_q)^{\frac{1}{q-1}} \left\|\nabla f(x^k)\right\|_{x^k}^{*\frac{q-2}{q-1}}$ for $M_q$-Hölder continuous function implies*

$$\frac{1}{2(1-\alpha_k)}\|\nabla f(y)\|_{x^k}^{*2}, \leq \left\langle \nabla f(y), \left[\nabla^2 f(x^k)\right]^{-1} \nabla f(x^k)\right\rangle \leq \|\nabla f(y)\|_{x^k}^* \left\|\nabla f(x^k)\right\|_{x^k}^*,$$

*which after rearranging yields $\|\nabla f(y)\|_{x^k}^* \leq 2(1-\alpha_k)\|\nabla f(x^k)\|_{x^k}^*$. Therefore if*

$$\alpha \leq \frac{1}{1 + (9M_q)^{\frac{1}{q-1}} \left\|\nabla f(x^k)\right\|_{x^k}^{*\frac{q-2}{q-1}}} \tag{33}$$

*or equivalently*

$$\overline{\alpha} \leq \left(1 + (9M_q)^{\frac{1}{q-1}} \left\|\nabla f(x^k)\right\|_{x^k}^{*\frac{q-2}{q-1}}\right)^{-1} \leq \left(1 + \sup_{q \in [2,4]} (9M_q)^{\frac{1}{q-1}} \left\|\nabla f(x^0)\right\|_{x^0}^{*\frac{q-2}{q-1}}\right)^{-1}. \tag{34}$$

*In such case, $\bar{c}$ can be set as $\bar{c} = 2(1 - \underline{\alpha})$.*

*Note that (34) is satisfied by smaller stepsizes, which damped Newton methods use globally until they converge to the neighborhood of the solution.*

**Remark** (Hölder continuity of Hessians). *For $L_{2,\nu}$-Hölder, Lemma 8 yields*

$$\|\nabla f(y)\|_{x^k}^* \leq \left(|1 - \alpha| + \frac{L_{2,\nu}}{1+\nu}\alpha^{1+\nu}\|\nabla f(x^k)\|_{x^k}^{*\nu}\right)\|\nabla f(x^k)\|_{x^k}^*, \tag{35}$$

*ensuring that without any limitation on $\overline{\alpha}$*

$$\bar{c}_x \stackrel{def}{=} \sup_{\alpha \in [\underline{\alpha}, \overline{\alpha}]} |1 - \alpha| + \frac{L_{2,\nu}}{1+\nu}\alpha^{1+\nu}\|\nabla f(x^k)\|_{x^k}^{*\nu} \tag{36}$$

$$= \max_{\alpha \in \{\underline{\alpha}, \overline{\alpha}, 1\}} |1 - \alpha| + \frac{L_{2,\nu}}{1+\nu}\alpha^{1+\nu}\|\nabla f(x^k)\|_{x^k}^{*\nu}. \tag{37}$$

*For $\underline{\alpha} \leftarrow 0, \overline{\alpha} \leftarrow 1$, we can set*

$$\bar{c} = \max\left\{1, \frac{L_{2,\nu}}{1+\nu}\|\nabla f(x^k)\|_{x^k}^{*\nu}\right\} \leq \max\left\{1, \frac{L_{2,\nu}}{1+\nu}\|\nabla f(x^0)\|_{x^0}^{*\nu}\right\}. \tag{38}$$

**Remark** ($L_{2,0}$-Hölder continuity). *For $L_{2,0}$-Hölder functions with $L_{2,0} \geq 1$, constant $\bar{c}$ simplifies to $\bar{c} \stackrel{def}{=} \overline{\alpha}\frac{L_{2,0}}{2} + |1 - \overline{\alpha}|$, because*

$$\begin{cases} \overline{\alpha}\left(\frac{L_{2,0}}{2} - 1\right) + 1 \geq \alpha\left(\frac{L_{2,0}}{2} - 1\right) + 1 \geq \frac{1}{2}, & \text{if } \alpha \leq 1, \\ \overline{\alpha}\left(\frac{L_{2,0}}{2} + 1\right) - 1 \geq \alpha\left(\frac{L_{2,0}}{2} + 1\right) - 1 \geq \frac{L_{2,0}}{2}, & \text{if } \alpha \geq 1. \end{cases} \tag{39}$$

## D  CONNECTION BETWEEN STEPSIZES AND REGULARIZATION

We show connections of particular stepsizes to regularized Newton methods. For fixed $\sigma > 0, \beta \geq 0$ define regularized model as

$$T_{\sigma,\beta}(x) \stackrel{\text{def}}{=} \underset{y \in \mathbb{R}^d}{\text{argmin}} \left\{ f(x) + \langle \nabla f(x), y - x \rangle + \frac{1}{2}\|y - x\|_x^2 + \frac{\sigma}{2 + \beta}\|y - x\|_x^{2+\beta} \right\}. \tag{40}$$

We can define optimization algorithm RN as

$$x^{k+1} \stackrel{\text{def}}{=} T_{\sigma,\beta}(x^k) \tag{41}$$

By first-order optimality condition, solution of model $h^* \stackrel{\text{def}}{=} T_{\sigma,\beta}(x) - x$ satisfy

$$\left(1 + \sigma\|h^*\|_x^\beta\right)\left[\nabla^2 f(x)\right] h^* = -\nabla f(x), \tag{42}$$

$$h^* = -\underbrace{\left(1 + \sigma\|h^*\|_x^\beta\right)^{-1}}_{\stackrel{\text{def}}{=} \alpha > 0}\left[\nabla^2 f(x)\right]^{-1}\nabla f(x). \tag{43}$$

Now iterates of RN are in the direction of Newton method (for any $\sigma$ and $\beta$) and we can write

$$h^* = -\alpha\left[\nabla^2 f(x)\right]^{-1}\nabla f(x), \tag{44}$$

$$\left[\nabla^2 f(x)\right] h^* = -\alpha\nabla f(x), \tag{45}$$

$$\|h^*\|_x = \alpha\|\nabla f(x)\|_x^*. \tag{46}$$

Substituting $\left[\nabla^2 f(x)\right] h^*$ back to the first-order optimality conditions we get

$$0 = \nabla f(x)\left(1 - \alpha - \alpha^{1+\beta}\sigma\|\nabla f(x)\|_x^{*\beta}\right). \tag{47}$$

Thus, $\alpha$ defined as a root of the polynomial

$$P[\alpha] \stackrel{\text{def}}{=} 1 - \alpha - \alpha^{1+\beta}\sigma\|\nabla f(x)\|_x^{*\beta} \tag{48}$$

satisfies first-order optimality condition. Note that $P[0] > 0$ and $P[1] \leq 0$, hence $P$ has root on interval $(0, 1]$. This will be the stepsize of our algorithm. Also note that P is monotone on $\mathbb{R}_+$,

$$P'[\alpha] = -1 - (1 + \beta)\alpha^\beta\sigma\|\nabla f(x)\|_x^{*\beta} < 0, \tag{49}$$

and consequently, the positive root of $P$ is unique.

## E  RELATIONS BETWEEN SMOOTHNESS CONSTANTS

First note that the parametrization $L_{p,\nu}$ is log-convex in $\nu$ and hence for $0 \leq \nu_1 \leq \nu \leq \nu_2 \leq 1$, it hold

$$L_{p,\nu} \leq [L_{p,\nu_1}]^{\frac{\nu_2 - \nu}{\nu_2 - \nu_1}}[L_{p,\nu_2}]^{\frac{\nu - \nu_1}{\nu_2 - \nu_1}}, \qquad \text{and} \qquad L_{p,\nu} \leq L_{p,0}^{1-\nu}L_{p,1}^\nu.$$

Consider any $\gamma \in [0, 1]$. From Hölders continuity, triangle inequality and definition of $L_{p,\nu}$,

$$\left\|\nabla^3 f(x)[y - x]\right\|_{op} \leq \left\|\nabla^2 f(x) - \nabla^2 f(y)\right\|_{op} + \frac{L_{3,\nu}}{1 + \nu}\|y - x\|_x^{1+\nu} \tag{50}$$

$$\leq L_{2,\gamma}\|x - y\|_x^\gamma + \frac{L_{3,\nu}}{1 + \nu}\|y - x\|_x^{1+\nu} \tag{51}$$

For $y \leftarrow x + \tau h$, where $\|h\|_x = 1, \tau > 0$, we can continue

$$\left\|\nabla^3 f(x)\right\|_{op} \leq \frac{L_{2,\gamma}}{\tau^{1-\gamma}} + \frac{L_{3,\nu}}{1 + \nu}\tau^\nu, \tag{52}$$

$$\leq \frac{2 + \nu}{1 + \nu}[L_{2,\gamma}]^{\frac{\nu}{1+\nu-\gamma}}\tau^{1-\gamma}[L_{3,\nu}]^{\frac{1}{1+\nu-\gamma}}, \qquad // \text{ by } \tau \leftarrow \left[\frac{L_{2,\gamma}}{L_{3,\nu}}\right]^{\frac{1}{1+\nu-\gamma}} \tag{53}$$

$$\leq \frac{3}{2}\sqrt{L_{2,0}L_{3,1}}, \qquad // \text{ by } \gamma \leftarrow 0, \nu \leftarrow 1 \tag{54}$$

and we can summarize

$$L_{3,0} = \sup_{x \neq y} \left\| \nabla^3 f(x) - \nabla^3 f(y) \right\|_{op} \leq \sup_{x \neq y} \left( \left\| \nabla^3 f(x) \right\|_{op} + \left\| \nabla^3 f(y) \right\|_{op} \right) \tag{55}$$

$$= 2 \sup_x \left\| \nabla^3 f(x) \right\|_{op} \leq \begin{cases} 2L_{2,1} \\ 3\sqrt{L_{2,0} L_{3,1}} \end{cases}. \tag{56}$$

**Lemma 6.** *If $L_{2,\nu}$ exists, for points $x^k, x^{k+1} = x^k - \alpha_k \left[ \nabla^2 f(x^k) \right]^{-1} \nabla f(x^k)$ holds decrease*

$$\left\| \nabla f(x^{k+1}) \right\|_{x^k}^* \leq \left( \theta_k + \frac{L_{2,\nu}}{1+\nu} \alpha_k^\nu \left\| \nabla f(x^k) \right\|_{x^k}^{*\nu} \right) \alpha_k \left\| \nabla f(x^k) \right\|_{x^k}^*,$$

*and hence, if $\nu > 0$ and $\theta_k \geq \left\| \nabla f(x^k) \right\|_{x^k}^{*\varepsilon}$ for $\varepsilon > 0$, and if the bound* (127) *exists (meaning that the Hessian does not change much), we have guaranteed superlinear local rate.*

**Remark.** *Hanzely et al. (2022) shows that $L_{2,1}$-Hölder continuity implies self-concordance, and (Nesterov, 2018, Theorem 4.1.3) proves that self-concordance implies positive definiteness of Hessian $\nabla^2 f$ the domain of function $f$ contains no straight line.*

## F  GENERALITY OF HIGHER-ORDER REGULARIZATION

In this section we explain how (7) encapsulates polynomial upper bounds $P[\|x - y\|_x]$ with smaller exponents. Writing regularization as a polynomial,

$$f(y) \leq \Phi_x(y) + P[\|x - y\|_x], \tag{57}$$

this can be bounded as

$$f(y) \leq \Phi_x(y) + A_1 + A_2 \|x - y\|_x^p, \tag{58}$$

where constants $A_1, A_2 > 0$ and degree $p$ are expressed in the lemma below. Notably, the next iterate $x^+$ set as the minimizer of the right-hand side of (58) is not affected by $A_1$, but the $A_1$ worsens guarantees on functional value decrease, $f(x^+) \leq f(x) + A_1$.

**Lemma 7.** *A polynomial $P$ with $d_P$ coefficients $a_k \geq 0$ and exponents $0 \leq b_1 \leq \cdots \leq b_{d_P}$,*

$$P[x] \overset{def}{=} \sum_{k=0}^{d_P} a_k x^{b_k},$$

*satisfies following bound with any $p \geq \max_{k \in \{1, \ldots, d_P\}} b_k$,*

$$P[x] \leq A_1 + A_2 x^p,$$

*where $A_1 = \frac{1}{p} \sum_{k=0}^{d_P} a_k (p - b_k), A_2 = \frac{1}{p} \sum_{k=0}^{d_P} a_k b_k$.*

**A surprising remark:** Similarly, we can replace even the quadratic term from Taylor polynomial, $\frac{1}{2} \|y - x\|_x^2$, by an upper bound in the form $A_1 + A_2 \|x - y\|_x^p$. This further simplifies the regularization and results in the Newton method with the **unbounded stepsize**

$$x^+ = x - \left( \frac{1}{(\sigma + 1) \|\nabla f(x^k)\|_{x^k}^{*\beta}} \right)^{\frac{1}{1+\beta}} \left[ \nabla^2 f(x) \right]^{-1} \nabla f(x).$$

As the gradient diminishes, the stepsize diverges to infinity. Yet, simultaneously, the functional value is guaranteed to not deteriorate by more than a constant factor.

*Proof of the remark.* We can bound the majorization as

$$T_{\sigma,\beta}(x) = \underset{y \in \mathbb{R}^d}{\arg\min} \left\{ f(x) + \langle \nabla f(x), y - x \rangle + \frac{1}{2} \|y - x\|_x^2 + \frac{\sigma}{2 + \beta} \|y - x\|_x^{2+\beta} \right\} \tag{59}$$

$$\leq \underset{y \in \mathbb{R}^d}{\arg\min} \left\{ f(x) + \langle \nabla f(x), y - x \rangle + \frac{\beta}{2(\beta + 2)} + \frac{\sigma + 1}{2 + \beta} \|y - x\|_x^{2+\beta} \right\} \tag{60}$$

$$= x - \left( \frac{1}{(\sigma + 1) \|\nabla f(x^k)\|_{x^k}^{*\beta}} \right)^{\frac{1}{1+\beta}} \left[ \nabla^2 f(x) \right]^{-1} \nabla f(x), \tag{61}$$

where stepsize was obtained as the positive root of polynomial

$$P[\alpha] \stackrel{\text{def}}{=} 1 - \alpha^{1+\beta}(\sigma+1)\|\nabla f(x^k)\|_{x^k}^{*\beta}.$$

$\square$

Surprisingly, stepsize is unbounded, and when $\|\nabla f(x)\|_x^* \to 0$, then $\alpha \to \infty$. This puzzling result has a simple explanation – such stepsize converges only to a neighborhood of the solution.

In practice, we could not observe stepsize larger than $5$ on any considered dataset. When close to the solution and the stepsize becomes larger than one, algorithm (61) stops converging closer to the solution, and functional values oscillate.

# G  ANALYSIS UNDER $s$-RELATIVE SIZE ASSUMPTION

In this section, we present global convergence guarantees under a novel characteristic called $s$-relative size recently proposed by Doikov et al. (2024).

Strict convexity implies $\beta_f(x,y) > 0$, we also have $\lim_{s \to \infty} D_s = D$, also $\frac{\beta_f(x,y)}{V_f} \le 1$, and

$$\langle \nabla f(x) - \nabla f(y), x - y \rangle \ge V_f \left( \frac{\|x-y\|_x}{D_s} \right)^s \tag{62}$$

Characteristic $D_s$ is log-convex function in $s$, and if $D_{s_1}, D_{s_2} < \infty$, then for $2 \le s_1 \le s \le s_2$ holds

$$D_s \le [D_{s_1}]^{\frac{s_2-s}{s_2-s_1}} [D_{s_2}]^{\frac{s-s_1}{s_2-s_1}}, \tag{63}$$

and $D_s$ is continuous on this segment.

**Remark.** *For self-concordant functions, it holds $\beta_f(x,y) \ge \|y-x\|_x^2$, and $D_s \le D^{1-\frac{2}{s}}V_f^{\frac{1}{s}}$.*

**Remark.** *For functions such that $\beta_f(x,y) \ge \mu_s\|x-y\|_x^s$ it holds $D_s \le \left(\frac{V_f}{\mu_s}\right)^{\frac{1}{s}}$. In particular, for self-concordant functions holds $\beta_f(x,y) \ge \|y-x\|_x^2$, and therefore $D_2 \le \sqrt{V_f}$.*

**Assumption 2.** *For some $s \ge 2$, value of $D_s$ is finite, $D_s < \infty$.*

**Lemma 8.** *For any $2 \le s \le q$, we have*

$$\left( \frac{D_q}{D} \right)^q \le \left( \frac{D_s}{D} \right)^s \tag{64}$$

*Proof of Lemma 8.* Analogical to Doikov et al. (2024). $\square$

Now for any $x, y \in \mathcal{Q}(x^0)$,

$$f(y) = f(x) + \langle \nabla f(x), y - x \rangle + \int_0^1 \frac{1}{\tau} \langle \nabla f(x + \tau(y-x)) - \nabla f(x), \tau(y-x) \rangle \, d\tau \tag{65}$$

$$\ge f(x) + \langle \nabla f(x), y - x \rangle + \frac{1}{s} V_f \left( \frac{\|x-y\|_x}{D_s} \right)^s, \tag{66}$$

and minimizing both sides w.r.t. $y$ independently, we get

$$\frac{s-1}{s} \left( \frac{D_s \|\nabla f(x)\|_x^*}{V_f} \right)^{\frac{s}{s-1}} \ge \frac{f(x) - f_*}{V_f} \tag{67}$$

Let us denote some constants that will appear in proofs.

$$\hat{\gamma} \stackrel{\text{def}}{=} \frac{q(s-1)}{(q-1)s} \in \left[ \frac{2}{3}, 2 \right], \qquad \text{and} \qquad 1 - \hat{\gamma} = \frac{q-s}{(q-1)s} \tag{68}$$

$$\omega_{q,s} \stackrel{\text{def}}{=} \frac{1}{2} \left( \frac{s}{s-1} \right)^{\hat{\gamma}} \left( \frac{V_f^{\frac{q}{s}}}{9M_q D_s^q} \right)^{\frac{1}{q-1}} = \frac{1}{2} \left( \frac{s}{s-1} \right)^{\frac{q(s-1)}{(q-1)s}} \left( \frac{V_f^{\frac{q}{s}}}{9M_q D_s^q} \right)^{\frac{1}{q-1}} \tag{69}$$

$$C_q \stackrel{\text{def}}{=} 2\gamma(q-1)(9M_q)^{\frac{1}{q-1}} D^{\frac{q}{q-1}} \tag{70}$$

Note that $\frac{\omega_{q,s}C_q}{\gamma(q-1)} = \left( \left( \frac{s}{s-1} \right)^{\frac{s-1}{s}} \frac{V_f^{\frac{1}{s}}D}{D_s} \right)^{\frac{q}{q-1}}$.

**Lemma 9.** *For $q \in [2,4]$ and $s \in [2,\infty)$, we have*

$$\frac{1}{(\hat{\gamma}-1)f_{k+1}^{\hat{\gamma}-1}} - \frac{1}{(\hat{\gamma}-1)f_k^{\hat{\gamma}-1}} \geq \omega_{q,s} \frac{\|\nabla f(x_{k+1})\|_{x_{k+1}}^{*2}}{\|\nabla f(x_k)\|_{x_k}^{*2}}. \tag{71}$$

*Proof.* Analogically to Doikov et al. (2024), denote $f_k \overset{\text{def}}{=} f(x^k) - f_*$.

$$f_k - f_{k+1} \overset{(14)}{\geq} \frac{1}{2} \left( \frac{1}{9M_q} \right)^{\frac{1}{q-1}} \frac{\|\nabla f(x^k)\|_{x^k}^{*2}}{\|\nabla f(x^k)\|_{x^k}^{*2}} \|\nabla f(x^k)\|_{x^k}^{*\frac{q}{q-1}} \tag{72}$$

$$\overset{(67)}{\geq} \frac{1}{2} \left( \frac{1}{9M_q} \right)^{\frac{1}{q-1}} \frac{\|\nabla f(x^{k+1})\|_{x^k}^{*2}}{\|\nabla f(x^k)\|_{x^k}^{*2}} \left( \frac{V_f^{\frac{1}{s}}}{D_s} \right)^{\frac{q}{q-1}} \left( \frac{s}{s-1} \right)^{\hat{\gamma}} f_k^{\hat{\gamma}} \tag{73}$$

$$= \frac{1}{2} \left( \frac{s}{s-1} \right)^{\hat{\gamma}} \left( \frac{V_f^{\frac{q}{s}}}{9M_q D_s^q} \right)^{\frac{1}{q-1}} \frac{\|\nabla f(x^{k+1})\|_{x^k}^{*2}}{\|\nabla f(x^k)\|_{x^k}^{*2}} f_k^{\hat{\gamma}} \tag{74}$$

$$= \omega_{q,s} \frac{\|\nabla f(x^{k+1})\|_{x^k}^{*2}}{\|\nabla f(x^k)\|_{x^k}^{*2}} f_k^{\hat{\gamma}}. \tag{75}$$

If $s \geq q$, then $\hat{\gamma} \in [1,2]$ and the function $y(x) \overset{\text{def}}{=} x^{\hat{\gamma}-1}$ is concave. With monotonicity of $\{f_k\}_{k \geq 0}$, we have

$$\frac{1}{(\hat{\gamma}-1)f_{k+1}^{\hat{\gamma}-1}} - \frac{1}{(\hat{\gamma}-1)f_k^{\hat{\gamma}-1}} = \frac{f_k^{\hat{\gamma}-1} - f_{k+1}^{\hat{\gamma}-1}}{(\hat{\gamma}-1)f_{k+1}^{\hat{\gamma}-1}f_k^{\hat{\gamma}-1}} \geq \frac{f_k - f_{k+1}}{f_{k+1}^{\hat{\gamma}-1}f_k} \geq \omega_{q,s} \frac{\|\nabla f(x_{k+1})\|_{x_k}^{*2}}{\|\nabla f(x_k)\|_{x_k}^{*2}}. \tag{76}$$

If $2 \leq s < q$, then $\hat{\gamma} < 1$ and the function $y(x) \overset{\text{def}}{=} x^{\hat{\gamma}-1}$ is concave. We have

$$\frac{1}{(\hat{\gamma}-1)f_{k+1}^{\hat{\gamma}-1}} - \frac{1}{(\hat{\gamma}-1)f_k^{\hat{\gamma}-1}} = \frac{f_k^{1-\hat{\gamma}} - f_{k+1}^{1-\hat{\gamma}}}{1-\hat{\gamma}} \geq \frac{f_k - f_{k+1}}{f_k^{\hat{\gamma}}} \geq \omega_{q,s} \frac{\|\nabla f(x_{k+1})\|_{x_k}^{*2}}{\|\nabla f(x_k)\|_{x_k}^{*2}}. \tag{77}$$

$\square$

**Theorem 8.** *Let function $f$ be $L_{p,\nu}$-Hölder continuous with finite $s$-relative size and $\gamma$-bounded Hessian change, $M_q, D_s < \infty$ for some $q \in [2,4]$ and $s \geq q$ and sequence of iterates $x^0, \ldots, x^k$ by generated by one of the algorithms RN, UN, GRLS. If all iterates had function suboptimality $f_k \overset{\text{def}}{=} f(x^k) - f_*$ worse than $\varepsilon > 0$, $f_t \geq \varepsilon$ for $t \in \{0, \ldots k\}$, then the algorithm did at most*

$$k \leq \frac{\gamma}{\omega_{q,s}(\hat{\gamma}-1)} \left[ \frac{1}{f_k^{\hat{\gamma}-1}} - \frac{1}{f_0^{\hat{\gamma}-1}} \right] + 2\ln \frac{\|\nabla f(x^0)\|_{x^0}^* D}{f_k} \tag{78}$$

$$\leq 2\gamma \frac{s(q-1)}{s-q} \left( \frac{s-1}{s} \right)^{\frac{q(s-1)}{(q-1)s}} \left( \frac{9M_q D_s^q}{V_f^{\frac{q}{s}}} \right)^{\frac{1}{q-1}} \left[ \varepsilon^{-\frac{s-q}{s(q-1)}} - f_0^{-\frac{s-q}{s(q-1)}} \right]$$

$$+ 2\ln \frac{\|\nabla f(x^0)\|_{x^0}^* D}{\varepsilon} \tag{79}$$

*steps. If $s = q$, treating RHS as limit together with $\lim_{a \to 0} \frac{b^{-a} - c^{-a}}{a} = \ln\left(\frac{c}{b}\right)$ guarantees the linear convergence rate*

$$k \le 2\gamma \frac{q-1}{q} \left(\frac{9 M_q D_q^q}{V_f}\right)^{\frac{1}{q-1}} \ln \frac{f_0}{\varepsilon} + 2 \ln \frac{\left\|\nabla f(x^0)\right\|_{x^0}^* D}{\varepsilon}. \tag{80}$$

**Remark.** *We can analogically guarantee the global linear convergence of Greedy Newton linesearch* GN *(18), but with a slightly different constant.*

*Proof.* Telescoping Lemma 9,

$$\frac{1}{(\hat{\gamma} - 1) f_k^{\hat{\gamma} - 1}} - \frac{1}{(\hat{\gamma} - 1) f_0^{\hat{\gamma} - 1}} \ge \omega_{q,s} \sum_{t=0}^{k-1} \frac{\left\|\nabla f(x^{t+1})\right\|_{x^t}^{*2}}{\left\|\nabla f(x^t)\right\|_{x^t}^{*2}} \tag{81}$$

$$\ge k \omega_{q,s} \left(\prod_{t=0}^{k-1} \frac{\left\|\nabla f(x^{t+1})\right\|_{x^t}^{*2}}{\left\|\nabla f(x^t)\right\|_{x^t}^{*2}}\right)^{\frac{1}{k}} \tag{82}$$

$$\ge \frac{k \omega_{q,s}}{\gamma} \left(\frac{f_k}{\left\|\nabla f(x^0)\right\|_{x^0}^* D}\right)^{\frac{k}{2}} \tag{83}$$

$$\ge \frac{k \omega_{q,s}}{\gamma} \exp\left(-\frac{2}{k} \ln \frac{\left\|\nabla f(x^0)\right\|_{x^0}^* D}{f_k}\right) \tag{84}$$

$$\ge \frac{k \omega_{q,s}}{\gamma} \left(1 - \frac{2}{k} \ln \frac{\left\|\nabla f(x^0)\right\|_{x^0}^* D}{f_k}\right) \tag{85}$$

$$= \frac{k \omega_{q,s}}{\gamma} - \frac{2 \omega_{q,s}}{\gamma} \ln \frac{\left\|\nabla f(x^0)\right\|_{x^0}^* D}{f_k}, \tag{86}$$

hence

$$k \le \frac{\gamma}{\omega_{q,s}(\hat{\gamma} - 1)} \left[\frac{1}{f_k^{\hat{\gamma} - 1}} - \frac{1}{f_0^{\hat{\gamma} - 1}}\right] + 2 \ln \frac{\left\|\nabla f(x^0)\right\|_{x^0}^* D}{f_k} \tag{87}$$

$$\le \frac{\gamma}{\omega_{q,s}(\hat{\gamma} - 1)} \left[\frac{1}{f_k^{\hat{\gamma} - 1}} - \frac{1}{f_0^{\hat{\gamma} - 1}}\right] + 2 \ln \frac{\left\|\nabla f(x^0)\right\|_{x^0}^* D}{\varepsilon}. \tag{88}$$

$$\square$$

**Theorem 9.** *Let funciton $f$ be $L_{p,\nu}$-Hölder continuous with finite $s$-relative size and $\gamma$-bounded Hessian change, $M_q, D_s < \infty$ for some $q \in [2, 4]$ and $2 \le s \le q$ and sequence of iterates $x^0, \ldots, x^k$ by generated by one of the algorithms* RN, UN, GRLS. *If all iterates were far from solution, $f_t \ge \varepsilon > 0$ and $g_t \stackrel{def}{=} \left\|\nabla f(x^t)\right\|_{x^t}^* \ge \delta > 0$ for $t \in \{0, \ldots k\}$, then the algorithm did at most*

$$k \le 2\gamma \frac{q}{s} \left(\frac{s-1}{s}\right)^{\frac{s-1}{q-1}} \left(\frac{9 M_q D_s^s D^{q-s}}{V_f}\right)^{\frac{1}{q-1}} \frac{s(q-1)}{q-s} \left[1 - \frac{s}{q} \left(\left(\frac{s}{s-1}\right)^{s-1} \frac{D_s^s}{V_f D^s} \varepsilon\right)^{\frac{q-s}{s(q-1)}}\right]$$

$$+ 2 \ln \frac{g_0}{\delta} \tag{89}$$

*steps. If $s = q$, treating RHS as a limit guarantees linear convergence rate*

$$k \le 2\gamma \frac{q-1}{q} \left(\frac{9 M_q D_q^q}{V_f}\right)^{\frac{1}{q-1}} \ln \left(\left(\frac{q}{q-1}\right)^{q-1} \frac{V_f D^q}{D_q^q \varepsilon}\right) + 2 \ln \frac{g_0}{\delta}. \tag{90}$$

*Proof.* Note $1 - \hat{\gamma} = \frac{q-s}{s(q-1)} > 0$. Let's split the analysis of the method into two stages, $k = m + n$. With $C_q = 2\gamma(q-1)(9M_q)^{\frac{1}{q-1}}D^{\frac{q}{q-1}}$, we bound the first stage,

$$C_q \frac{1}{f_m^{\frac{1}{q-1}}} \geq C_q \left[ \frac{1}{f_m^{\frac{1}{q-1}}} - \frac{1}{f_0^{\frac{1}{q-1}}} \right] \overset{(121)}{\geq} m \left( \frac{g_m}{g_0} \right)^{\frac{2}{m}} = m \exp\left( \frac{2}{m} \ln \frac{g_m}{g_0} \right) \tag{91}$$

$$\geq m + 2\ln \frac{g_m}{g_0} = m + 2\ln \frac{g_m}{\delta} - 2\ln \frac{g_0}{\delta}. \tag{92}$$

For the second stage, telescoping inequalities for $t = m, \dots, k-1$

$$\frac{1}{\omega_{q,s}(1-\hat{\gamma})} \left[ f_{t+1}^{1-\hat{\gamma}} - f_t^{1-\hat{\gamma}} \right] \geq \frac{\|\nabla f(x_{t+1})\|_{x_{t+1}}^{*2}}{\|\nabla f(x_t)\|_{x_t}^{*2}}, \tag{93}$$

we get

$$\frac{\gamma}{\omega_{q,s}(1-\hat{\gamma})} \left[ f_m^{1-\hat{\gamma}} - \varepsilon^{1-\hat{\gamma}} \right] \geq \gamma \sum_{t=m}^{k-1} \frac{\|\nabla f(x_{t+1})\|_{x_{t+1}}^{*2}}{\|\nabla f(x_t)\|_{x_t}^{*2}} \geq n \left( \frac{g_k}{g_m} \right)^{\frac{2}{n}} \geq n \left( \frac{\delta}{g_m} \right)^{\frac{2}{n}} \tag{94}$$

$$\geq n - 2\ln \frac{g_m}{\delta}. \tag{95}$$

Expressing $n, m$ from the inequalities above and adding them together yields

$$k \leq C_q \frac{1}{f_m^{\frac{1}{q-1}}} + \frac{\gamma}{\omega_{q,s}(1-\hat{\gamma})} \left[ f_m^{1-\hat{\gamma}} - \varepsilon^{1-\hat{\gamma}} \right] + 2\ln \frac{g_0}{\delta}. \tag{96}$$

Note that $1 - \hat{\gamma} = \frac{q-s}{s(q-1)}$. Minimizer of RHS in $f_m$ is achieved at

$$f_m^* \overset{\text{def}}{=} \left( \frac{C_q \omega_{q,s}}{\gamma(q-1)} \right)^{\frac{s(q-1)}{q}} = \left( \frac{s}{s-1} \right)^{\frac{s}{s-1}} \frac{V_f D^s}{D_s^s}. \tag{97}$$

Substituting definitions of $f_m^*, \omega_{q,s}, C_q, \hat{\gamma}$ into the terms we get

$$C_q \frac{1}{f_m^{*\frac{1}{q-1}}} = 2\gamma(q-1) \left( \frac{s-1}{s} \right)^{\frac{s-1}{q-1}} \left( \frac{9M_q D_s^s D^{q-s}}{V_f} \right)^{\frac{1}{q-1}},$$

$$\frac{\gamma}{\omega_{q,s}(1-\hat{\gamma})} f_m^{*(1-\hat{\gamma})} = \gamma \frac{s(q-1)}{q-s} \frac{1}{\omega_{q,s}} f_m^{*\frac{q-s}{s(q-1)}}$$

$$= 2\gamma \frac{s(q-1)}{q-s} \left( \frac{s-1}{s} \right)^{\frac{s-1}{q-1}} \left( \frac{9M_q D_s^s D^{q-s}}{V_f} \right)^{\frac{1}{q-1}},$$

$$\frac{\gamma}{\omega_{q,s}(1-\hat{\gamma})} \varepsilon^{1-\hat{\gamma}} = 2\gamma \frac{s(q-1)}{q-s} \left( \frac{s-1}{s} \right)^{\frac{q(s-1)}{(q-1)s}} \left( \frac{9M_q D_s^q}{V_f^{\frac{q}{s}}} \right)^{\frac{1}{q-1}} \varepsilon^{\frac{q-s}{s(q-1)}},$$

and plugging them back in, we conclude

$$k \leq C_q \frac{1}{f_m^{*\frac{1}{q-1}}} + \frac{\gamma}{\omega_{q,s}(1-\hat{\gamma})} \left[ f_m^{*(1-\hat{\gamma})} - \varepsilon^{1-\hat{\gamma}} \right] + 2\ln \frac{g_0}{\delta}$$

$$= 2\gamma(q-1) \frac{q}{q-s} \left( \frac{s-1}{s} \right)^{\frac{s-1}{q-1}} \left( \frac{9M_q D_s^s D^{q-s}}{V_f} \right)^{\frac{1}{q-1}} - \frac{\gamma}{\omega_{q,s}(1-\hat{\gamma})} \varepsilon^{1-\hat{\gamma}} + 2\ln \frac{g_0}{\delta}$$

$$= 2\gamma \frac{q}{s} \left( \frac{s-1}{s} \right)^{\frac{s-1}{q-1}} \left( \frac{9M_q D_s^s D^{q-s}}{V_f} \right)^{\frac{1}{q-1}} \frac{s(q-1)}{q-s} \times$$

$$\times \left[ 1 - \frac{s}{q} \left( \left( \frac{s}{s-1} \right)^{s-1} \frac{V_f D^s}{D_s^s} \right)^{\frac{q-s}{s(q-1)}} \varepsilon^{\frac{q-s}{s(q-1)}} \right] + 2\ln \frac{g_0}{\delta}.$$

$\square$

# H PROOFS

## H.1 PROOF OF LEMMA 7

*Proof of Lemma 7.* Using weighed AG inequality, for $0 \leq b \leq p$, we have

$$x^b \leq \frac{(p-b) + bx^p}{p}. \tag{98}$$

We use this inequality for each term of the polynomial. □

## H.2 PROOF OF PROPOSITION 1

*Proof of Proposition 1.* We can derive all of the inequalities straightforwardly

$$\nabla f(y) - \nabla f(x) - \nabla^2 f(x)\,[y-x] = \int_0^1 \left(\nabla^2 f(x + \tau(y-x)) - \nabla^2 f(x)\right)[y-x]d\tau$$

$$\left\|\nabla f(y) - \nabla f(x) - \nabla^2 f(x)\,[y-x]\right\|_x^* \leq \int_0^1 \left\|\nabla^2 f(x + \tau(y-x)) - \nabla^2 f(x)\right\|_{op}\|y-x\|_x d\tau$$

$$\leq L_{2,\nu}\|y-x\|_x^{1+\nu}\int_0^1 \tau^\nu d\tau$$

$$= \frac{L_{2,\nu}}{1+\nu}\|y-x\|_x^{1+\nu},$$

$$\nabla^2 f(y) - \nabla^2 f(x) - \nabla^3 f(x)\,[y-x] = \int_0^1 \left(\nabla^3 f(x + \tau(y-x)) - \nabla^3 f(x)\right)[y-x]d\tau$$

$$\left\|\nabla^2 f(y) - \nabla^2 f(x) - \nabla^3 f(x)\,[y-x]\right\|_{op} \leq \int_0^1 \left\|\nabla^3 f(x + \tau(y-x)) - \nabla^3 f(x)\right\|_{op}\|y-x\|_x d\tau$$

$$\leq L_{3,\nu}\|y-x\|_x^{1+\nu}\int_0^1 \tau^\nu d\tau$$

$$= \frac{L_{3,\nu}}{1+\nu}\|y-x\|_x^{1+\nu},$$

$$\nabla f(y) - \nabla f(x) - \nabla^2 f(x)\,[y-x] - \frac{1}{2}\nabla^3 f(x)[y-x]^2$$

$$= \int_0^1 \int_0^\tau \left(\nabla^3 f(x + \sigma(y-x)) - \nabla^3 f(x)\right)[y-x]^2 d\sigma d\tau$$

$$\left\|\nabla f(y) - \nabla f(x) - \nabla^2 f(x)\,[y-x] - \frac{1}{2}\nabla^3 f(x)[y-x]^2\right\|_x^*$$

$$\leq \int_0^1 \int_0^\tau \left\|\nabla^3 f(x + \sigma(y-x)) - \nabla^3 f(x)\right\|_x^*\|y-x\|_x^2 d\sigma d\tau$$

$$\leq L_{3,\nu}\|y-x\|_x^{2+\nu}\int_0^1 \int_0^\tau \sigma^\nu d\sigma d\tau$$

$$= \frac{L_{3,\nu}}{(1+\nu)(2+\nu)}\|y-x\|_x^{2+\nu}.$$

□

## H.3 PROOF OF LEMMA 1

*Proof of Lemma 1.* For any $x, h, y \in \mathbb{E}$ and taking $y = x + \tau u$ for $\tau > 0$, $\|u\|_x = 1$

$$0 \leq \|h\|_y^2 \leq \|h\|_x^2 + \left\langle \nabla^3 f(x)[h]^2, y - x \right\rangle + \frac{L_{3,\nu}}{1+\nu}\|y-x\|_x^{1+\nu}\|h\|_x^2$$

$$0 \leq \frac{1}{\tau}\|h\|_x^2 + \left\langle \nabla^3 f(x)[h]^2, u \right\rangle + \frac{L_{3,\nu}\tau^\nu}{1+\nu}\|h\|_x^2$$

$$\left\|\nabla^3 f(x)[h]^2\right\|_x^* \leq \left(\frac{1}{\tau} + \frac{L_{3,\nu}\tau^\nu}{1+\nu}\right)\|h\|_x^2$$

Setting

$$\tau = \left(\frac{1+\nu}{L_{3,\nu}}\right)^{\frac{1}{1+\nu}},$$

we get

$$\left\|\nabla^3 f(x)[h]^2\right\|_x^* \leq 2\left(\frac{L_{3,\nu}}{1+\nu}\right)^{\frac{1}{1+\nu}}\|h\|_x^2.$$

Setting $x^k = x, h = x^{k+1} - x^k$ we get

$$\left\|\nabla^3 f(x^k)[x^{k+1} - x^k]^2\right\|_{x^k}^* \leq 2\left(\frac{L_{3,\nu}}{1+\nu}\right)^{\frac{1}{1+\nu}}\left\|x^{k+1} - x^k\right\|_{x^k}^2 = 2\left(\frac{L_{3,\nu}}{1+\nu}\right)^{\frac{1}{1+\nu}}\alpha_k^2\left\|\nabla f(x^k)\right\|_{x^k}^{*2}$$

$\square$

## H.4 PROOF OF LEMMA 6

*Proof.* Proof of Lemma 6.

$$
\begin{aligned}
\left\|\nabla f(x^{k+1})\right\|_{x^k}^* &= \left\|\nabla f(x^{k+1}) - \nabla^2 f(x^k)\left[x^{k+1} - x^k\right] - \alpha_k \nabla f(x^k)\right\|_{x^k}^* \\
&= \left\|\nabla f(x^{k+1}) - \nabla f(x^k) - \nabla^2 f(x^k)\left[x^{k+1} - x^k\right] + (1-\alpha_k)\nabla f(x^k)\right\|_{x^k}^* \\
&\leq \left\|\nabla f(x^{k+1}) - \nabla f(x^k) - \nabla^2 f(x^k)\left[x^{k+1} - x^k\right]\right\|_{x^k}^* + (1-\alpha_k)\left\|\nabla f(x^k)\right\|_{x^k}^* \\
&\leq \frac{L_{2,\nu}}{1+\nu}\left\|x^{k+1} - x^k\right\|_{x^k}^{1+\nu} + (1-\alpha_k)\left\|\nabla f(x^k)\right\|_{x^k}^* \qquad \text{(if } L_{2,\nu} \text{ exists)} \\
&= \frac{L_{2,\nu}}{1+\nu}\alpha_k^{1+\nu}\left\|\nabla f(x^k)\right\|_{x^k}^{*(1+\nu)} + (1-\alpha_k)\left\|\nabla f(x^k)\right\|_{x^k}^* \\
&= \left(1-\alpha_k + \frac{L_{2,\nu}}{1+\nu}\alpha_k^{1+\nu}\left\|\nabla f(x^k)\right\|_{x^k}^{*\nu}\right)\left\|\nabla f(x^k)\right\|_{x^k}^* \\
&= \left(\theta_k + \frac{L_{2,\nu}}{1+\nu}\alpha_k^\nu\left\|\nabla f(x^k)\right\|_{x^k}^{*\nu}\right)\alpha_k\left\|\nabla f(x^k)\right\|_{x^k}^*.
\end{aligned}
$$

Hence

$$\left\|\nabla f(x^{k+1})\right\|_{x^k}^* \leq \begin{cases} 2\frac{L_{2,\nu}}{1+\nu}\alpha_k^{1+\nu}\left\|\nabla f(x^k)\right\|_{x^k}^{*(1+\nu)} & \text{if } \theta_k \leq \frac{L_{2,\nu}}{1+\nu}\alpha_k^\nu\left\|\nabla f(x^k)\right\|_{x^k}^{*\nu} \\ 2\theta_k\alpha_k\left\|\nabla f(x^k)\right\|_{x^k}^* & \text{if } \theta_k \geq \frac{L_{2,\nu}}{1+\nu}\alpha_k^\nu\left\|\nabla f(x^k)\right\|_{x^k}^{*\nu} \end{cases}$$

$\square$

## H.5 PROOF OF LEMMA 4

We provide separate proofs for cases $p = 2$ and $p = 3$.

*Proof of Lemma 4, case $p = 2$.* We can rewrite the Hölder continuity for points $x^k, x^{k+1}$ s.t. $x^{k+1} = x^k - \alpha_k \left( \nabla^2 f(x^k) \right)^{-1} \nabla f(x^k)$

$$
\left( \frac{L_{2,\nu}}{1+\nu} \left( \alpha_k \| \nabla f(x^k) \|_{x^k}^* \right)^{1+\nu} \right)^2
$$
$$
= \left( \frac{L_{2,\nu}}{1+\nu} \| x^{k+1} - x^k \|_{x^k}^{1+\nu} \right)^2
$$
$$
\geq \left\| \nabla f(x^{k+1}) - \nabla f(x^k) - \nabla^2 f(x^k) \left[ x^{k+1} - x^k \right] \right\|_{x^k}^{*2}
$$
$$
= \left\| \nabla f(x^{k+1}) - \nabla f(x^k) + \alpha_k \nabla f(x^k) \right\|_{x^k}^{*2}
$$
$$
= \left\| \nabla f(x^{k+1}) - (1 - \alpha_k) \nabla f(x^k) \right\|_{x^k}^{*2}
$$
$$
= \left\| \nabla f(x^{k+1}) \right\|_{x^k}^{*2} + (1 - \alpha_k)^2 \left\| \nabla f(x^k) \right\|_{x^k}^{*2} - 2 (1 - \alpha_k) \left\langle \nabla f(x^{k+1}), \left[ \nabla^2 f(x^k) \right]^{-1} \nabla f(x^k) \right\rangle.
$$

We are going to set $\sigma$ so that

$$
\frac{1 - \alpha_k}{2} \| \nabla f(x^k) \|_{x^k}^{*2} \geq \frac{1}{2(1-\alpha_k)} \left( \frac{L_{2,\nu}}{1+\nu} \left( \alpha_k \| \nabla f(x^k) \|_{x^k}^* \right)^{1+\nu} \right)^2, \tag{99}
$$

and hence, we can conclude the proof by rearranging,

$$
\left\langle \nabla f(x^{k+1}), \left[ \nabla^2 f(x^k) \right]^{-1} \nabla f(x^k) \right\rangle
$$
$$
\geq \frac{1}{2(1-\alpha_k)} \| \nabla f(x^{k+1}) \|_{x^k}^{*2} + \frac{1 - \alpha_k}{2} \| \nabla f(x^k) \|_{x^k}^{*2} - \frac{1}{2(1-\alpha_k)} \left( \frac{L_{2,\nu}}{1+\nu} \left( \alpha_k \| \nabla f(x^k) \|_{x^k}^* \right)^{1+\nu} \right)^2
$$
$$
\geq \frac{1}{2(1-\alpha_k)} \| \nabla f(x^{k+1}) \|_{x^k}^{*2}.
$$

Now we are going to choose $\sigma$ to satisfy (99). Because $\alpha_k$ is a root of a polynomial $P$, we have

$$
1 - \alpha_k - \alpha_k^{1+\beta} \lambda_k = 0,
$$

so the equation (99) is equivalent to

$$
1 - \alpha_k = \alpha_k^{1+\beta} \lambda_k \geq \frac{L_{2,\nu}}{1+\nu} \alpha_k^{1+\nu} \| \nabla f(x^k) \|_{x^k}^{*\nu},
$$
$$
\theta_k \geq \frac{L_{2,\nu}}{1+\nu} \alpha_k^{\nu} \| \nabla f(x^k) \|_{x^k}^{*\nu}.
$$

$\square$

*Proof of Lemma 4, case $p = 3$.* We can rewrite the Hölder continuity for points $x^k, x^{k+1}$ s.t. $x^{k+1} = x^k - \alpha_k \left( \nabla^2 f(x^k) \right)^{-1} \nabla f(x^k)$

$$
\frac{L_{3,\nu}}{(1+\nu)(2+\nu)} \left( \alpha_k \| \nabla f(x^k) \|_{x^k}^* \right)^{2+\nu} \tag{100}
$$
$$
= \frac{L_{3,\nu}}{(1+\nu)(2+\nu)} \| x^{k+1} - x^k \|_{x^k}^{2+\nu} \tag{101}
$$
$$
\geq \left\| \nabla f(x^{k+1}) - \nabla f(x^k) - \nabla^2 f(x^k)[x^{k+1} - x^k] - \frac{1}{2} \nabla^3 f(x^k)[x^{k+1} - x^k]^2 \right\|_{x^k}^* \tag{102}
$$
$$
= \left\| \nabla f(x^{k+1}) - (1 - \alpha_k) \nabla f(x^k) - \frac{1}{2} \nabla^3 f(x^k)[x^{k+1} - x^k]^2 \right\|_{x^k}^*. \tag{103}
$$

Squaring, then using Chauchy-Schwartz inequality twice and then, lastly, Lemma 1

$$\left(\frac{L_{3,\nu}}{(1+\nu)(1+\nu)}\left(\alpha_k\|\nabla f(x^k)\|_{x^k}^*\right)^{2+\nu}\right)^2$$

$$\geq \left\|\nabla f(x^{k+1}) - (1-\alpha_k)\nabla f(x^k) - \frac{1}{2}\nabla^3 f(x^k)[x^{k+1}-x^k]^2\right\|_{x^k}^{*2}$$

$$= \|\nabla f(x^{k+1})\|_{x^k}^{*2} + (1-\alpha_k)^2\|\nabla f(x^k)\|_{x^k}^{*2} + \frac{1}{4}\|\nabla^3 f(x^k)[x^{k+1}-x^k]^2\|_{x^k}^{*2}$$

$$-2(1-\alpha_k)\left\langle\nabla f(x^{k+1}), \left[\nabla^2 f(x^k)\right]^{-1}\nabla f(x^k)\right\rangle$$

$$+(1-\alpha_k)\left\langle\left[\nabla^2 f(x^k)\right]^{-\frac{1}{2}}\nabla f(x^k), \left[\nabla^2 f(x^k)\right]^{-\frac{1}{2}}\nabla^3 f(x^k)[x^{k+1}-x^k]^2\right\rangle$$

$$-\left\langle\left[\nabla^2 f(x^k)\right]^{-\frac{1}{2}}\nabla f(x^{k+1}), \left[\nabla^2 f(x^k)\right]^{-\frac{1}{2}}\nabla^3 f(x^k)[x^{k+1}-x^k]^2\right\rangle$$

$$\geq \frac{1}{2}\|\nabla f(x^{k+1})\|_{x^k}^{*2} + (1-\alpha_k)^2\|\nabla f(x^k)\|_{x^k}^{*2} - \frac{1}{4}\|\nabla^3 f(x^k)[x^{k+1}-x^k]^2\|_{x^k}^{*2}$$

$$-2(1-\alpha_k)\left\langle\nabla f(x^{k+1}), \left[\nabla^2 f(x^k)\right]^{-1}\nabla f(x^k)\right\rangle$$

$$-(1-\alpha_k)\|\nabla f(x^k)\|_{x^k}^*\|\nabla^3 f(x^k)[x^{k+1}-x^k]^2\|_{x^k}$$

$$\geq \frac{1}{2}\|\nabla f(x^{k+1})\|_{x^k}^{*2} + (1-\alpha_k)^2\|\nabla f(x^k)\|_{x^k}^{*2} - \left(\frac{L_{3,\nu}}{1+\nu}\right)^{\frac{2}{1+\nu}}\alpha_k^4\|\nabla f(x^k)\|_{x^k}^4$$

$$-2(1-\alpha_k)\left\langle\nabla f(x^{k+1}), \left[\nabla^2 f(x^k)\right]^{-1}\nabla f(x^k)\right\rangle$$

$$-2\left(\frac{L_{3,\nu}}{1+\nu}\right)^{\frac{1}{1+\nu}}\alpha_k^2(1-\alpha_k)\|\nabla f(x^k)\|_{x^k}^{*3}.$$

Rearranging yields

$$\left\langle\nabla f(x^{k+1}), \left[\nabla^2 f(x^k)\right]^{-1}\nabla f(x^k)\right\rangle$$

$$\geq \frac{1}{4(1-\alpha_k)}\|\nabla f(x^{k+1})\|_{x^k}^{*2} + \frac{1-\alpha_k}{2}\|\nabla f(x^k)\|_{x^k}^{*2} - \frac{1}{2}\left(\frac{L_{3,\nu}}{1+\nu}\right)^{\frac{2}{1+\nu}}\frac{\alpha_k^4}{1-\alpha_k}\|\nabla f(x^k)\|_{x^k}^{*4}$$

$$-\left(\frac{L_{3,\nu}}{1+\nu}\right)^{\frac{1}{1+\nu}}\alpha_k^2\|\nabla f(x^k)\|_{x^k}^{*3} - \frac{1}{2(1-\alpha_k)}\left(\frac{L_{3,\nu}}{(1+\nu)(2+\nu)}\right)^2\left(\alpha_k\|\nabla f(x^k)\|_{x^k}^*\right)^{2(2+\nu)}.$$

Finally, we are going to set $\theta_k$ so that

$$\frac{1-\alpha_k}{6}\|\nabla f(x^k)\|_{x^k}^{*2} \geq \frac{1}{2}\left(\frac{L_{3,\nu}}{1+\nu}\right)^{\frac{2}{1+\nu}}\frac{\alpha_k^4}{1-\alpha_k}\|\nabla f(x^k)\|_{x^k}^{*4} \tag{104}$$

$$\frac{1-\alpha_k}{6}\|\nabla f(x^k)\|_{x^k}^{*2} \geq \left(\frac{L_{3,\nu}}{1+\nu}\right)^{\frac{1}{1+\nu}}\alpha_k^2\|\nabla f(x^k)\|_{x^k}^{*3} \tag{105}$$

$$\frac{1-\alpha_k}{6}\|\nabla f(x^k)\|_{x^k}^{*2} \geq \frac{1}{2(1-\alpha_k)}\left(\frac{L_{3,\nu}}{(1+\nu)(2+\nu)}\right)^2\left(\alpha_k\|\nabla f(x^k)\|_{x^k}^*\right)^{2(2+\nu)} \tag{106}$$

and then we can conclude

$$\left\langle\nabla f(x^{k+1}), \left[\nabla^2 f(x^k)\right]^{-1}\nabla f(x^k)\right\rangle \geq \frac{1}{4(1-\alpha_k)}\|\nabla f(x^{k+1})\|_{x^k}^{*2}.$$

Note that the choice of stepsize implies

$$1-\alpha_k = \alpha_k^{1+\beta}\lambda_k$$

and (104), (105), (106) are satisfied as

$$1 - \alpha_k = \alpha_k^{1+\beta}\lambda_k \geq$$

$$\begin{cases} \sqrt{3}\left(\frac{L_{3,\nu}}{1+\nu}\right)^{\frac{1}{1+\nu}}\alpha_k^2\left\|\nabla f(x^k)\right\|_{x^k}^* & \text{if } \theta_k \geq \sqrt{3}\left(\frac{L_{3,\nu}}{1+\nu}\right)^{\frac{1}{1+\nu}}\alpha_k\left\|\nabla f(x^k)\right\|_{x^k}^* \\ 6\left(\frac{L_{3,\nu}}{1+\nu}\right)^{\frac{1}{1+\nu}}\alpha_k^2\left\|\nabla f(x^k)\right\|_{x^k}^* & \text{if } \theta_k \geq 6\left(\frac{L_{3,\nu}}{1+\nu}\right)^{\frac{1}{1+\nu}}\alpha_k\left\|\nabla f(x^k)\right\|_{x^k}^* \\ \frac{\sqrt{3}L_{3,\nu}}{(1+\nu)(1+\nu)}\alpha_k^{2+\nu}\left\|\nabla f(x^k)\right\|_{x^k}^{*(1+\nu)} & \text{if } \theta_k \geq \frac{\sqrt{3}L_{3,\nu}}{(1+\nu)(2+\nu)}\alpha_k^{1+\nu}\left\|\nabla f(x^k)\right\|_{x^k}^{*(1+\nu)}. \end{cases}$$

We can ensure (104), (105), (106) by

$$\theta_k \geq \alpha_k\left\|\nabla f(x^k)\right\|_{x^k}^* \max\left\{6\left(\frac{L_{3,\nu}}{1+\nu}\right)^{\frac{1}{1+\nu}}, \frac{\sqrt{3}L_{3,\nu}}{(1+\nu)(2+\nu)}\alpha_k^\nu\left\|\nabla f(x^k)\right\|_{x^k}^{*\nu}\right\}.$$

$\square$

## H.6 Towards the proof of Theorem 2

We unify cases $p = 2, 3$ with the Lemma 5.

**Corollary 3.** *Lemma 5 with $\gamma = \nu$ implies that choice $\theta_k = \left(\frac{L_{2,\nu}}{1+\nu}\right)^{\frac{1}{1+\nu}}\left\|\nabla f(x^k)\right\|_{x^k}^{*\frac{\nu}{1+\nu}}$ satisfies $\theta_k$ requirement of Lemma 4 for $p = 2$ and therefore it implies decrease as Doikov et al. (2024),*

$$f(x^k) - f(x^{k+1}) \geq \frac{1}{\theta_k}\left\|\nabla f(x^{k+1})\right\|_{x^k}^{*2} \geq \left(\frac{1+\nu}{L_{2,\nu}}\right)^{\frac{1}{1+\nu}}\frac{\left\|\nabla f(x^{k+1})\right\|_{x^k}^{*2}}{\left\|\nabla f(x^k)\right\|_{x^k}^{*\frac{\nu}{1+\nu}}}. \tag{107}$$

*Lemma 5 with $\gamma \in \{1, 1+\nu\}$ implies that the choice*

$$\theta_k \geq$$

$$\left\|\nabla f(x^k)\right\|_{x^k}^{*\frac{1}{2}}\max\left\{\left(\frac{6^{1+\nu}L_{3,\nu}}{1+\nu}\right)^{\frac{1}{2(1+\nu)}}, \left(\frac{\sqrt{3}L_{3,\nu}}{(1+\nu)(2+\nu)}\right)^{\frac{1}{2+\nu}}\left\|\nabla f(x^k)\right\|_{x^k}^{*\frac{\nu}{2(2+\nu)}}\right\}, \tag{108}$$

*satisfies (22), and therefore Lemma 4 for $p = 3$ implies decrease*

$$f(x^k) - f(x^{k+1}) \geq \frac{1}{2\theta_k}\left\|\nabla f(x^{k+1})\right\|_{x^k}^{*2} \tag{109}$$

$$\geq \frac{1}{\max\left\{\left(\frac{6^{1+\nu}L_{3,\nu}}{1+\nu}\right)^{\frac{1}{2(1+\nu)}}, \left(\frac{\sqrt{3}L_{3,\nu}}{(1+\nu)(2+\nu)}\right)^{\frac{1}{2+\nu}}\left\|\nabla f(x^k)\right\|_{x^k}^{*\frac{\nu}{2(2+\nu)}}\right\}}\frac{\left\|\nabla f(x^{k+1})\right\|_{x^k}^{*2}}{\left\|\nabla f(x^k)\right\|_{x^k}^{*\frac{1}{2}}}. \tag{110}$$

*On the other hand, choice of $\theta_k = \left(\frac{6^{1+\nu}L_{3,\nu}}{1+\nu}\right)^{\frac{1}{2+\nu}}\left\|\nabla f(x^k)\right\|_{x^k}^{*\frac{1+\nu}{2+\nu}}$ in Lemma 4 ($p = 3$ case) implies decrease as Doikov et al. (2024),*

$$f(x^k) - f(x^{k+1}) \geq \frac{1}{2\theta_k}\left\|\nabla f(x^{k+1})\right\|_{x^k}^{*2} \geq \frac{1}{2}\left(\frac{1+\nu}{6^{1+\nu}L_{3,\nu}}\right)^{\frac{1}{2+\nu}}\frac{\left\|\nabla f(x^{k+1})\right\|_{x^k}^{*2}}{\left\|\nabla f(x^k)\right\|_{x^k}^{*\frac{1+\nu}{2+\nu}}}. \tag{111}$$

### H.6.1 Proof of Theorem 2

We can combine previous corollaries.

*Proof of Theorem 2.* For $p = 2$, choice $\theta_k = \left(\frac{L_{p,\nu}}{p-1+\nu}\right)^{\frac{1}{p-1+\nu}}\left\|\nabla f(x^k)\right\|_{x^k}^{*\frac{p-2+\nu}{p-1+\nu}}$ implies

$$f(x^k) - f(x^{k+1}) \geq \left(\frac{p-1+\nu}{L_{p,\nu}}\right)^{\frac{1}{p-1+\nu}}\frac{\left\|\nabla f(x^{k+1})\right\|_{x^k}^{*2}}{\left\|\nabla f(x^k)\right\|_{x^k}^{*\frac{p-2+\nu}{p-1+\nu}}}. \tag{112}$$

For $p = 3$, choice $\theta_k = 6 \left( \frac{L_{p,\nu}}{3(p-1+\nu)} \right)^{\frac{1}{p-1+\nu}} \left\| \nabla f(x^k) \right\|_{x^k}^{* \frac{p-2+\nu}{p-1+\nu}}$ implies

$$f(x^k) - f(x^{k+1}) \geq \frac{1}{12} \left( \frac{3(p-1+\nu)}{L_{p,\nu}} \right)^{\frac{1}{p-1+\nu}} \frac{\left\| \nabla f(x^{k+1}) \right\|_{x^k}^{*2}}{\left\| \nabla f(x^k) \right\|_{x^k}^{* \frac{p-2+\nu}{p-1+\nu}}}. \tag{113}$$

And for any $p \in \{2, 3\}$ we have that $\theta_k = 6 \left( \frac{L_{p,\nu}}{3(p-1+\nu)} \right)^{\frac{1}{p-1+\nu}} \left\| \nabla f(x^k) \right\|_{x^k}^{* \frac{p-2+\nu}{p-1+\nu}}$ implies

$$f(x^k) - f(x^{k+1}) \geq \frac{1}{12} \left( \frac{3(p-1+\nu)}{L_{p,\nu}} \right)^{\frac{1}{p-1+\nu}} \frac{\left\| \nabla f(x^{k+1}) \right\|_{x^k}^{*2}}{\left\| \nabla f(x^k) \right\|_{x^k}^{* \frac{p-2+\nu}{p-1+\nu}}}. \tag{114}$$

$\square$

## H.7 PROOF OF LEMMA 5

*Proof of Lemma 5.* Consider any $c_2, \delta > 0$. Inequality $\theta_k \geq c_2^{\frac{1}{1+\delta}}$ implies

$$\frac{1}{\theta_k{}^\delta} c_2 \geq c_2 \alpha_k^\delta,$$

which is ensured by

$$\theta_k \geq \frac{1}{\theta_k{}^\delta} c_2,$$

or equivalently

$$\theta_k \geq c_2^{\frac{1}{1+\delta}}.$$

Now, choice $c_2 = c_3 \left\| \nabla f(x^k) \right\|_{x^k}^{*\delta}$ guarantees that $\theta_k \geq c_3^{\frac{1}{1+\delta}} \left\| \nabla f(x^k) \right\|_{x^k}^{*\frac{\delta}{1+\delta}}$ ensures $\theta_k \geq c_3 \left( \alpha_k \left\| \nabla f(x^k) \right\|_{x^k}^* \right)^\delta$. $\square$

## H.8 PROOF OF COROLLARY 3

*Proof of Corollary 3.* For the first part of (22), we use $\alpha_k, \nu \in [0, 1]$ to bound $\frac{1}{\theta_k^{\frac{1}{1+\nu}}} \geq \alpha_k^{\frac{1}{1+\nu}} \geq \alpha_k$ and

$$\frac{1}{\theta_k^{\frac{1}{1+\nu}}} 6 \left( \frac{L_{3,\nu}}{1+\nu} \right)^{\frac{1}{1+\nu}} \left\| \nabla f(x^k) \right\|_{x^k}^* \geq 6 \left( \frac{L_{3,\nu}}{1+\nu} \right)^{\frac{1}{1+\nu}} \alpha_k \left\| \nabla f(x^k) \right\|_{x^k}^*.$$

Now, the first part of (22) is ensured by $\theta_k$ so that

$$\theta_k \geq \frac{1}{\theta_k^{\frac{1}{1+\nu}}} 6 \left( \frac{L_{3,\nu}}{1+\nu} \right)^{\frac{1}{1+\nu}} \left\| \nabla f(x^k) \right\|_{x^k}^*,$$

or equivalently

$$\theta_k \geq \left( \frac{6^{1+\nu} L_{3,\nu}}{1+\nu} \right)^{\frac{1}{2+\nu}} \left\| \nabla f(x^k) \right\|_{x^k}^{* \frac{1+\nu}{2+\nu}}.$$

We ensure the second part of (22) directly using Lemma 5 and together with first part we have

$$\theta_k \geq \max \left\{ \left( \frac{6^{1+\nu} L_{3,\nu}}{1+\nu} \right)^{\frac{1}{2+\nu}} \left\| \nabla f(x^k) \right\|_{x^k}^{* \frac{1+\nu}{2+\nu}}, \left( \frac{\sqrt{3} L_{3,\nu}}{(1+\nu)(2+\nu)} \right)^{\frac{1}{2+\nu}} \left\| \nabla f(x^k) \right\|_{x^k}^{* \frac{1+\nu}{2+\nu}} \right\}$$

$$= \left( \frac{L_{3,\nu}}{1+\nu} \right)^{\frac{1}{2+\nu}} \left\| \nabla f(x^k) \right\|_{x^k}^{* \frac{1+\nu}{2+\nu}} \max \left\{ 6^{\frac{1+\nu}{2+\nu}}, \left( \frac{\sqrt{3}}{2+\nu} \right)^{\frac{1}{2+\nu}} \right\}$$

$$= \left( \frac{6^{1+\nu} L_{3,\nu}}{1+\nu} \right)^{\frac{1}{2+\nu}} \left\| \nabla f(x^k) \right\|_{x^k}^{* \frac{1+\nu}{2+\nu}}.$$

$\square$

## H.9 PROOF OF LEMMA 2

*Proof of Lemma 2.* For $0 \leq \beta \leq 1$, function $y(x) = x^\beta, x \geq 0$ is concave, which implies

$$a^\beta - b^\beta \geq \frac{\beta}{a^{1-\beta}}(a - b), \quad \forall a > b \geq 0, \tag{115}$$

which we will be using for $\beta \overset{\text{def}}{=} \frac{1}{q-1} = (0,1]$. We rewrite functional value decrease with $f_k \overset{\text{def}}{=} f(x^k) - f_*$ as

$$\frac{1}{f_{k+1}^\beta} - \frac{1}{f_k^\beta} = \frac{f_k^\beta - f_{k+1}^\beta}{f_k^\beta f_{k+1}^\beta} \overset{(115)}{\geq} \frac{\beta(f_k - f_{k+1})}{f_k f_{k+1}^\beta} \overset{(14)}{\geq} \beta c_5 \frac{\left\|\nabla f(x^{k+1})\right\|_{x^k}^{*2}}{\left\|\nabla f(x^k)\right\|_{x^k}^{*\frac{q-2}{q-1}}} \frac{1}{f_k f_{k+1}^{\frac{1}{q-1}}} \tag{116}$$

$$\geq \beta c_5 \frac{\left\|\nabla f(x^{k+1})\right\|_{x^k}^{*2}}{\left\|\nabla f(x^k)\right\|_{x^k}^{*(2-\frac{q}{q-1})}} \frac{1}{f_k^{\frac{q}{q-1}}} \geq \frac{\beta c_5}{D^{1+\beta}} \frac{\left\|\nabla f(x^{k+1})\right\|_{x^k}^{*2}}{\left\|\nabla f(x^k)\right\|_{x^k}^{*2}}, \tag{117}$$

where in the last step we used the convexity of $f$ in the form $f_k \leq D\left\|\nabla f(x^k)\right\|_{x^k}^*$. We can continue by summing it for $k = 0, \ldots, n - 1$,

$$\frac{1}{f_n^\beta} - \frac{1}{f_0^\beta} \geq \frac{\beta c_5}{D^{1+\beta}} \sum_{k=0}^{n-1} \frac{\left\|\nabla f(x^{k+1})\right\|_{x^k}^{*2}}{\left\|\nabla f(x^k)\right\|_{x^k}^{*2}} \tag{118}$$

$$\overset{AG}{\geq} \frac{\beta c_5 n}{D^{1+\beta}} \left(\prod_{k=0}^{n-1} \frac{\left\|\nabla f(x^{k+1})\right\|_{x^k}^{*2}}{\left\|\nabla f(x^k)\right\|_{x^k}^{*2}}\right)^{\frac{1}{n}} \tag{119}$$

$$= \frac{\beta c_5 n}{D^{1+\beta}} \left(\prod_{k=1}^{n-1} \frac{\left\|\nabla f(x^k)\right\|_{x^{k-1}}^{*2}}{\left\|\nabla f(x^k)\right\|_{x^k}^{*2}}\right)^{\frac{1}{n}} \left(\frac{\left\|\nabla f(x^n)\right\|_{x^{n-1}}^*}{\left\|\nabla f(x^0)\right\|_{x^0}^*}\right)^{\frac{2}{n}} \tag{120}$$

$$\geq \frac{\gamma \beta c_5 n}{D^{1+\beta}} \left(\frac{f_n}{\left\|\nabla f(x^0)\right\|_{x^0}^* D}\right)^{\frac{2}{n}} \tag{121}$$

$$= \frac{\gamma \beta c_5 n}{D^{1+\beta}} \exp\left(-\frac{2}{n} \ln\left(\frac{\left\|\nabla f(x^0)\right\|_{x^0}^* D}{f_n}\right)\right) \tag{122}$$

$$\geq \frac{\gamma \beta c_5 n}{D^{1+\beta}} \left(1 - \frac{2}{n} \ln\left(\frac{\left\|\nabla f(x^0)\right\|_{x^0}^* D}{f_n}\right)\right) \tag{123}$$

We can bound $f_n$ based on the size of $\frac{2}{n} \frac{\left\|\nabla f(x^0)\right\|_{x^0}^* D}{f_n}$.

1. If $\frac{2}{n} \ln\left(\frac{\left\|\nabla f(x^0)\right\|_{x^0}^* D}{f_n}\right) \geq \frac{1}{2}$, then $f_n \leq \left\|\nabla f(x^0)\right\|_{x^0}^* D \exp\left(-\frac{k}{4}\right)$.

2. If $\frac{2}{n} \ln\left(\frac{\left\|\nabla f(x^0)\right\|_{x^0}^* D}{f_n}\right) < \frac{1}{2}$, then

$$\frac{1}{f_n^\beta} > \frac{1}{f_n^\beta} - \frac{1}{f_0^\beta} \geq \frac{\gamma \beta c_5 n}{2D^{1+\beta}} \Leftrightarrow f_n < \left(\frac{2D^{1+\beta}}{\gamma \beta c_5 n}\right)^{\frac{1}{\beta}} = \frac{D^q \left(2(q-1)\right)^{q-1}}{\gamma^{q-1} c_5^{q-1} n^{q-1}} \tag{124}$$

Hence

$$f_n \leq \frac{D^q \left(2(q-1)\right)^{q-1}}{\gamma^{q-1} c_5^{q-1} n^{q-1}} + \left\|\nabla f(x^0)\right\|_{x^0}^* D \exp\left(-\frac{k}{4}\right). \tag{125}$$

$\square$

### H.10 Proof of Theorem 3

*Proof of Theorem 3.* Cauchy-Schwartz inequality together with condition (13) in Theorem 2 imply inequality

$$\left\|\nabla f(x^{k+1})\right\|_{x^k}^* \left\|\nabla f(x^k)\right\|_{x^k}^* \geq \left\langle \nabla f(x^{k+1}), \left[\nabla^2 f(x^k)\right]^{-1} \nabla f(x^k)\right\rangle \geq \frac{1}{2\alpha_k \theta_k}\left\|\nabla f(x^{k+1})\right\|_{x^k}^{*2},$$

(126)

which together with bounded Hessian change assumption yields

$$\left\|\nabla f(x^k)\right\|_{x^k}^* \geq \frac{1}{2\alpha_k \theta_k}\left\|\nabla f(x^{k+1})\right\|_{x^k}^* \geq \frac{\gamma}{2\alpha_k \theta_k}\left\|\nabla f(x^{k+1})\right\|_{x^{k+1}}^* \geq \frac{\gamma}{2\theta_k}\left\|\nabla f(x^{k+1})\right\|_{x^{k+1}}^*.$$

(127)

This for $\theta_k$ from (12) guarantees local superlinear rate for $q > 2$. $\qquad\square$

### H.11 Proof of Theorem 4

*Proof of Theorem 4.* Theorem 2 implies that Algorithm 1 satisfies requirements of Lemma 2 with correspondent $q$ and $c_5 = \frac{1}{2}\left(\frac{1}{9M_q}\right)^{\frac{1}{q-1}}$. The convergence rate follows. $\qquad\square$

### H.12 Proof of Lemma 3

*Proof of Lemma 3.* We will prove the statement by induction. The base for $\sigma_0$ holds. For $k$-th iteration, consider 2 cases based on the number of iterations of the inner loop.

1. Algorithm 2 continues after $j_k > 0$ inner iterations. Note that if $\theta_{k,j_k-1}$ satisfied (12), Theorem 2 guarantees the continuation condition to be satisfied for $j_k - 1$. Consequently, $\theta_{k,j_k-1}$ does not satisfy (12) for any $q \in [2,4]$, and hence

$$\sigma_{k+1} = \frac{\theta_{k,j_k-1}}{\left\|\nabla f(x^k)\right\|_{x^k}^{*\beta}} < \inf_{q \in [2,4]}(9M_q)^{\frac{1}{q-1}}\left\|\nabla f(x^k)\right\|_{x^k}^{*\frac{q-2}{q-1}-\beta} = \mathcal{H}\left(x^k\right).$$

(128)

2. Algorithm continues after $j = 0$ iterates, then from (127) we have

$$\sigma_{k+1} = \frac{\sigma_k}{\rho} \leq \frac{1}{\rho}\mathcal{H}\left(x^{k-1}\right) \leq \frac{1}{\rho\gamma^{\frac{q-2}{q-1}}}\mathcal{H}\left(x^k\right) \leq \mathcal{H}\left(x^k\right).$$

(129)

For the total number of oracle calls $N_K$,

$$N_K = \sum_{k=0}^{K-1}(1 + j_k) = K + \sum_{k=0}^{K-1}\log_\rho \frac{c\sigma_{k+1}}{\sigma_k} = 2K + \log_\rho \frac{\sigma_K}{\sigma_0}$$

(130)

$$\leq 2K + \log_\rho \frac{\mathcal{H}\left(\left\|x^{k-1}\right\|_{x^{k-1}}^*\right)}{\sigma_0}.$$

(131)

$\qquad\square$

### H.13 Proof of Theorem 5

*Proof of Theorem 5.* Algorithm 2 sets $x^{k+1} = x_{j_k}^k$ so that

$$\left\langle \nabla f(x_{j_k-1}^k), n^k\right\rangle < \frac{1}{2\alpha_{k,j_k-1}\theta_{k,j_k-1}}\left\|\nabla f(x_{j_k-1}^k)\right\|_{x^k}^{*2},$$

(132)

$$\left\langle \nabla f(x_{j_k}^k), n^k\right\rangle \geq \frac{1}{2\alpha_{k,j_k}\theta_{k,j_k}}\left\|\nabla f(x_{j_k}^k)\right\|_{x^k}^{*2}.$$

(133)

From Theorem 2 we can see that while $\theta_{k,j_{k-1}} = \theta_{k,j_k}/\rho$ does not satisfy (13) for any $q \in [2,4]$ and $\theta_{k,j_k}$ satisfies (12) for some $q$, therefore

$$\theta_{k,j_k} \geq (9M_q)^{\frac{1}{q-1}} \left\| \nabla f(x^k) \right\|_{x^k}^{* \frac{q-2}{q-1}} \qquad \exists q \in [2,4] \tag{134}$$

$$\theta_{k,j_k} < \rho \, (9M_q)^{\frac{1}{q-1}} \left\| \nabla f(x^k) \right\|_{x^k}^{* \frac{q-2}{q-1}} \qquad \forall q \in [2,4] \tag{135}$$

$$\theta_{k,j_k} < \rho \inf_{q \in [2,4]} (9M_q)^{\frac{1}{q-1}} \left\| \nabla f(x^k) \right\|_{x^k}^{* \frac{q-2}{q-1}}, \tag{136}$$

hence estimate $\theta_{k,j_k}$ is at most constant $\rho$ times worse than any plausible parametrization of $(q, M_q)$, and therefore, even the best plausible parametrization. In particular, for

$$q^* \stackrel{\text{def}}{=} \operatorname*{argmin}_{q \in [2,4]} 9M_q D \left( \frac{4\rho^2 D(q-1)}{k} \right)^{q-1} + \left\| \nabla f(x^0) \right\|_{x^0}^* D \exp\left( -\frac{k}{4} \right), \tag{137}$$

we have that from Theorem 2

$$f(x^k) - f(x^{k+1}) \geq \frac{\rho}{2} \left( \frac{1}{9M_{q^*}} \right)^{\frac{1}{q^*-1}} \frac{\left\| \nabla f(x^{k+1}) \right\|_{x^k}^{*2}}{\left\| \nabla f(x^k) \right\|_{x^k}^{* \frac{q^*-2}{q^*-1}}}. \tag{138}$$

The rest of the proof is analogous to the proof of Theorem 4. $\qquad \square$

