# OpenReview forum: "Newton Method Revisited: Global Convergence Rates up to $O(1/k^3)$  for Stepsize Schedules and Linesearch Procedures"
_ICLR.cc/2026/Conference — ICLR 2026 Poster_

### Official Review · Reviewer_rUdG · 2025-10-27

**Soundness:** 4
**Presentation:** 3
**Contribution:** 3
**Rating:** 6
**Confidence:** 4

**Summary:**

This paper presents new convergence guarantees for stepsized Newton methods under Holder continuity assumptions on the Hessian or third derivatives. The authors reinterpret the classical Newton method as a third-order tensor method and propose a family of stepsize schedules (RN), as well as linesearch and backtracking variants (GRLS, UN). These achieve global rates up to O(k^-3), improving upon the best-known O(k^2) results. The paper also provides convergence results for the “Greedy Newton” linesearch, supported by clear theoretical analysis and numerical experiments.

**Strengths:**

1. The theoretical contributions are solid and clearly positioned relative to prior work (notably Hanzely et al., Doikov et al., and Nesterov’s tensor methods).

2. The paper successfully extends the analysis of Newton-type methods into the regime of Holder continuous third derivatives, showing rates that were previously thought unattainable for basic Newton variants.

3. The exposition is mathematically rigorous and detailed, with clear structure and self-contained proofs.

4. The proposed algorithms (RN, UN, GRLS) are simple, practical, and connect naturally to existing Newton variants.

5.  The paper propose variants with adaptive sptesizes, which is important since we cannot

6. Experimental results are well designed and demonstrate that the proposed methods are competitive, often outperforming baselines.

**Weaknesses:**

1. The experimental section, though helpful, remains somewhat limited in scope (mostly synthetic or classical problems). Stronger empirical validation on large-scale or nonconvex deep learning tasks would better support the practical relevance.

2. The proposed linesearch requires evaluating the dual norm, which in turn involves an additional matrix inversion at each iteration. This cost is not discussed, and it can be significant in large-scale settings.

3. The experimental section reports convergence in terms of iterations (steps) rather than the actual number of matrix inversions or Hessian factorizations, which makes the practical comparison somewhat misleading.

4. The paper’s tone occasionally overstates its novelty. Conceptually, the main contribution builds on a straightforward observation: by substituting the Newton step into the Hölder continuity bound and optimizing the resulting inequality, one obtains improved global rates. This is an elegant and technically solid development, but the underlying idea remains relatively simple rather than transformative. The paper's strength lies in its clarity and rigor, not in a fundamentally new algorithmic principle.

5. The claim that the method can be viewed as a “third-order” method feels overstated. The algorithm still uses only first and second derivatives; the “third-order” aspect comes only from the smoothness assumption and the resulting analysis. This is somewhat analogous to calling gradient descent a “second-order” method because it relies on a bound on the Hessian variation.

**Questions:**

See weaknesses.

---

> ### Author Response · Authors · 2025-11-26
> **Rebuttal**
>
> ## Response to Reviewer rUdG
>
> We thank Reviewer rUdG for the positive evaluation and for the helpful suggestions, particularly regarding empirical scope and the discussion of computational costs. We are happy to clarify any of these points further.
>
> ### Experimental scope and nonconvex large-scale tasks
>
> We agree that additional experiments on large-scale or deep learning tasks would further support the practical relevance. However, such settings are typically dominated by first-order methods; second-order methods are known to often overfit and to be computationally prohibitive for large neural networks. Since our focus is on the step-size theory for Newton-type methods (rather than scalable approximations), it is outside of the scope of the project.
>
> ### Cost of dual norms and matrix inversions
>
> We agree that evaluating dual norms using inverse Hessians can be expensive in large-scale scenarios. However, this cost primarily affects GRLS, which we do **not** recommend as a practical algorithm.  (We proposed it for 3 reasons: demonstrating the meaningfulness of the one-step decrease formula, motivating the practical UN backtracking procedure, and connecting our analysis to classical Greedy Newton line search.)
>
> For UN, the cost is essentially that of a standard Newton method plus a logarithmic number of backtracking evaluations; it does not require additional matrix inversions beyond those needed for the Newton step itself. We will clarify this distinction.
>
> ### Iteration counts vs computational cost
>
> We reported convergence primarily in terms of iteration counts, as this cleanly reflects algorithmic progress and makes theoretical comparisons transparent. For the simpler step-size variants, Figures 2 and 3 also report convergence in wall-clock time. In the revision, we will expand this discussion and state explicitly which computational costs are counted (e.g., Hessian factorizations/inversions), to make the practical implications clearer.
>
> ### Tone and claims about novelty
>
> We appreciate the remark and will adjust the tone where appropriate. The core mechanism is indeed conceptually simple. Our main contribution lies in showing that this yields $\mathcal{O}(k^{-3})$ rates for stepsize schedules for Newton methods (simple schedule RN as well as adaptive variant UN).
>
> ### “Third-order method” terminology
>
> We agree that calling the method “third-order” can be misleading, since only first and second derivatives are evaluated. We will consistently refer to it as a “second-order method with third-order smoothness assumptions.”

---

> > ### Author Response · Authors · 2025-12-01
> > **Response follow-up**
> >
> > Dear reviewer,
> > We would like to kindly ask you to take another look at our rebuttal response together with the updated manuscript, and, if the revisions satisfactorily resolve your comments, consider revising the score.

---

### Official Review · Reviewer_T7B3 · 2025-10-28

**Soundness:** 2
**Presentation:** 2
**Contribution:** 3
**Rating:** 4
**Confidence:** 4

**Summary:**

The paper studies various choices of step-size for Newton's method and the convergence guarantees they provide. It generalizes recent ideas (notably from Hanzely et al. 2022) that exploits the link between specific step-size choices and third-order regularization techniques. The main generalization revolves around higher-order regularity assumptions on the objective functions, considering in particular third-order H√∂lder continuity.

The paper provides various algorithms, the main ones called RN, when the smoothness parameters of the function are known, and UN when they are unknown. The authors provide various convergence results for functions that have bounded level sets (which is equivalent to coercivity for convex functions). The results essentially extend known results for twice differentiable functions to the case of 3rd order differentiable functions and considering any order $\nu\in[0,1]$ of Holder continuity (when often only the continuous case $\nu=1$ is considered).

The contribution is good and interesting, yet the writing is, in my opinion, problematic at many points, which make the paper and the importance of various results unclear (see below), or sometimes misleading (for example I do not believe the paper improves the known results for the case $p=2, \nu=1$, yet this is implictely implied in Sec. 1.5, see hereafter).
Additionnally, I have several important concerns regarding the practical usefulness of the main algorithms (RN, UN and GRLS) introduced in the paper, which I detail in the weaknesses section.


Overall, the paper has merits and the contribution is fair and of interest to the community. Yet I belive that the paper needs significant improvements the writing and to give stronger arguments to convince the reader on the  the interest and impact of the main algorithms and results. It seems to me that it is yet to demonstrate how useful they are in practice compared to existing Newton's methods.
I would welcome clarifications and feedback from the authors on these key points. I believe the paper is not ready for publication in its current state but I would increase my rating if the writing is improved to clarify several key points as recommended below.

**Strengths:**

* Strong theoretical results on several new variants of Newton's method.
* Thorough analysis discussing all the main problems with Newton-like methods (line search or not, unknown regularization, etc.).
* Generalization of previous results to higher-order smoothness and H√∂lder continuity.
* Clear explanation of prior work
* Overall rigor on the theoretical side (despite some clarifications needed in the writting, especially in the introduction section).
* The RN algorithm is interesting and rather simple to implement when the smoothness constant is known. I have concerns regarding the other methods but RN is promissing.

**Weaknesses:**

* The introduction would be difficult to follow for reader not familiar with Newton's methods are many things are discussed without being properly defined (or often defined a few pages later). For example, Newton's methods is introduced in equation (1) without a step-size, but this step-size is then extensively discussed.
* The original motivation given by the authors in the introduction was to improve the known rate for Newton's methods in the case $p=2$ and to bring it closer to the optimal rate. Yet the main contribution is in the end for $p=3$, as in the case $p=2$ the results do not improve the $O(k^{-2})$ previously known rate.
* Some results would benefit from being further discussed to assert their usefulness in practice, a few examples are given in the questions below.
* The GRLS method in Section 3 seems much more expensive that the Greedy Newton method, since it requires computing $\langle \nabla f(y),\left(\nabla^2 f(x_k)\right)^{-1}\nabla f(y)\rangle$, possibly for several $y$ at each iteration.
The authors then explain the method empirically produces step-sizes close to those of Greedy Newton. Therefore I am not sure that the method has a practical interest due to its high computational cost compared to (GN)?
* The Universal Newton (UN) method is meant to be used on function where the smoothness constant $M_q$ is unknown, but it requires the value of $\gamma$ which is also related to the smoothness of the problem and is likely to be unknown as well. Similarly (see the questions below), it is not clear how $\theta_{k,0}$ is initialized sufficiently small (and it is not stated in the algorithm), since the true value is unknown. Therefore I am not sure that UN is more easy to use in practice than RN.

**Questions:**

* In Section 1.1, what does "basic" methods means? Is there a rigorous definition of it?
* Could the author clarify the global assumptions made on $f$ throughout the paper? Indeed, in Sec. 1.2, $f$ is convex twice differentiable and has non-degenerate Hessian. Since Holder continuity implies continuity, we later guess that it is actually $C^2$, or is this not correct?
* The assumption in Sec 2.2 is equivalent to coercivity (because $f$ is convex), so adding the $C^2$ property and non-degeneracy discussed above, it seems that $f$ is actually strongly convex. If this is correct this should be more straightforwardly stated.
* Similarly, the authors denote $x^\star$ any element of $\argmin_x f(x)$ but is this argmin assumed to be non-empty? Could the author clarify this sentence?
* The rate obtained for $p=2$, $\nu=1$ (so Holder-continuity of 2nd derivative) is no better than $O(k^{-2})$, so the current work does not improve on previous work for this case. Improvement require further assumptions. Yet the contribution section 1.5 makes it sound as it the case. I recommend reformulating or clarifying if I missed something, in the current form the contribution statement is misleading.
* In Theorem 3, is the factor of the superlinear local rate always smaller than $1$? That is, is it guaranteed that $\frac{2}{\gamma} (9M_q)^{1/q-1}<1$? I recommend further discussing this result and its consequences.
* In the case where smoothness parameters are unknown, the proposed Universal Newton (UN) method resorts to doing a line-search on $\theta_k$ to then use a step-size $\alpha_k = \frac{1}{1+\theta_k}$. Could the author clarify whether there are real advantages compared to  doing the line-search directly on $\alpha_k$ as usually done?
* In Section 4, regarding the UN algorithm, the authors say that the backtracking strategy consists in starting with an estimate of $\theta_k$ that is smaller than the true value. How is it possible to ensure this since the true value is unknown?

---

> ### Author Response · Authors · 2025-11-26
> **Rebuttal**
>
> We thank Reviewer T7B3 for the thorough analysis, constructive criticism, and suggestions on both theory and presentation. We are happy to clarify or refine any of these points further.
>
> ### GRLS line search vs Greedy Newton
>
> We agree that GRLS is significantly more expensive than Greedy Newton and is not meant as a practical algorithm. Its role in the paper is:
>
> 1. to demonstrate the usefulness of the one-step decrease formula,
> 2. to motivate the practical UN backtracking procedure, and
> 3. to connect our analysis to the classical Greedy Newton line search.
>
> We will clarify this explicitly.
>
> ### Dependence of UN on constants
>
> Regarding the Universal Newton (UN) method:
>
> - The backtracking **scaling factor** is written as $\gamma$ (to match Assumption 1), but it can be replaced by *any* constant $C>1$ (the theory does not require knowing the $\gamma$).
> - The initial smoothness estimate $\sigma_0$ can be chosen arbitrarily small. If $\sigma_0 < \sigma$, at most $O(\log(\sigma))$ backtracking iterations are needed to reach a valid range.
> - If $\sigma_0 > \sigma$, the algorithm will still be converging with smaller stepsizes (similarly to the overestimation of smoothness in other methods).
> - A practical trick we had used is to *decrease* $\sigma_k$ between iterations (line 11 of UN, -1 in exponent), which lets $\sigma_k$ shrink toward the local smoothness constant without significant overhead.
>
> Thus, UN does not require precise knowledge of problem-specific constants to be usable.
>
> ### Meaning of “basic methods”
>
> By “basic methods” (Section 1.1), we informally mean methods that use a single sequence of iterates, do not rely on internal subsolvers or nested optimization problems, and do not maintain additional memory (such as quasi-Newton curvature updates).
> We can clarify it explicitly.
>
> ### Clarifications on assumptions and notation
> We apologize for the confusion; we aimed to list function properties instead of referring to a notation potentially unknown to the reader. We clarify it for easier readability.
>
> ### Improvement in the Hölder-continuous Hessian case
>
> For $p=2, \nu=1$ (Hölder-continuous Hessian with exponent 1), our rate $\mathcal{O}(k^{-2})$ matches, but does not improve upon, known results. The improvement is in all other combinations of $p$ and $\nu$,  **general Hölder regime**, with even $\mathcal{O}(k^{-3})$ rate when third derivatives are Hölder continuous $p=3, \nu=1$. We will rewrite Section 1.5 to avoid any implication that the $\beta=1$ case is improved.
>
> ### Theorem 3 and the local superlinear rate
>
> The factor in Theorem 3 is not globally less than 1; otherwise it would imply *global* superlinear convergence. Superlinear convergence holds in a neighborhood where
> $$
> \frac{2}{\gamma}(9M)^{\frac{1}{q-1}}\Vert \nabla f(x^k)\Vert_{x^k}^{\ast(2 - \frac{1}{q-1})}
> \le
> \Vert \nabla f(x^k)\Vert_{x^k}^{\ast},
> $$
>
> or equivalently,
>
> $$
> \Vert \nabla f(x^k)\Vert_{x^k}^{*}
> \le
> \left(\frac{\gamma}{2}\right)^{\frac{q-1}{q-2}}
> \frac{1}{(9M)^{1/(q-2)}}.
> $$
>
> We will make this neighborhood condition explicit and briefly discuss its consequences.
>
>
> ### UN vs direct linesearch on the stepsize
>
> There are indeed many ways to perform linesearch directly on the stepsize (e.g., Armijo, Wolfe, Goldstein). To our knowledge, UN is the **first adaptive Newton linesearch with non-asymptotic global convergence guarantees**. Traditional linesearch rules are primarily heuristic in this context. In our experiments, UN performed competitively with the traditional linesearch criteria; a more exhaustive empirical comparison is part of our ongoing research.
>
> ### Initialization of $\sigma_0$ when $\sigma$ is unknown
>
> As mentioned above, the algorithm only requires choosing a positive $\sigma_0$. If $\sigma_0$ is too small, backtracking increases it in at most $O(\log(\sigma))$ steps until the sufficient decrease condition holds, as stated in Lemma 3. No prior knowledge of the true $\sigma$ is needed. We will make this more explicit in the algorithm description.

---

> > ### Comment · Reviewer_T7B3 · 2025-11-27
> >
> > Thank you for answering my concerns. Most of them were about clarifications on the actual contribution, the assumptions and the practical usefulness of the proposed algorithms. I agree with what the authors have answered, so I have no further questions. I am now waiting to see these changes incorporated in the new version before revising my rating.

---

> > > ### Author Response · Authors · 2025-12-01
> > > **Response follow-up**
> > >
> > > We have updated the paper to address the reviewers’ concerns as outlined in our rebuttal response. We kindly invite you to review the updated version to verify that the contributions, assumptions, and usefulness of our algorithms are now clearly presented.

---

### Official Review · Reviewer_tDN2 · 2025-10-31

**Soundness:** 3
**Presentation:** 3
**Contribution:** 3
**Rating:** 6
**Confidence:** 3

**Summary:**

This paper investigates globalization strategies for the Newton method under Hölder-type smoothness assumptions, including Hölder continuity of Hessians and third-order derivatives. The authors make three key contributions: (i) reinterpreting stepsize-controlled Newton methods as higher-order (third-order) regularized methods; (ii) proposing a new class of stepsize schedules (Root Newton — RN) along with corresponding line-search and backtracking procedures (GRLS and UN) that do not require exact knowledge of smoothness constants; and (iii) establishing global convergence rates up to $\mathcal{O}(k^{-3})$ under appropriate Hölder conditions. The work also provides local superlinear and linear convergence guarantees under additional assumptions, supported by numerical experiments comparing RN/UN/GRLS against existing Newton globalization techniques. The manuscript positions its main contribution as improving the best-known global rate for Newton stepsize methods from $\mathcal{O}(k^{-2})$ to $\mathcal{O}(k^{-3})$ by leveraging Hölder continuity of higher-order derivatives.

**Strengths:**

1. The paper achieves an improved global convergence rate of $\mathcal{O}(k^{-3})$ under Hölder continuity of higher-order derivatives, advancing the state-of-the-art for stepsize-based Newton methods.
2. The proposed GRLS and UN procedures eliminate the need for known smoothness constants while retaining theoretical guarantees, enhancing practical applicability.
3. A comprehensive set of convergence results is established, including local superlinear, global linear (under additional assumptions), and global sublinear rates.
4. Experimental results validate the theoretical claims and demonstrate the effectiveness of the proposed methods.

**Weaknesses:**

1. On Page 3, Line 132, the authors claim that the RN stepsize schedule generalizes the AICN stepschedule when $p=2$, $\nu=1$ (i.e., $q=3$). However, substituting these parameters yields $$\alpha_k = \frac{1}{1 + (9L_{2,1})^{1/2} \|\nabla f(x^k)\|_{x^k}^{*\frac12}},$$

which differs structurally from the AICN stepsize $$\alpha_k = \frac{2}{1 + \sqrt{1 + 2\sigma \|\nabla f(x^k)\|_{x^k}^*}},$$ even up to a constant factor. This claim requires clarification or correction.

2. On Page 6, Line 320, the authors state that Assumption 1 has been used in [1], but this assumption does not appear explicitly in the referenced work.

3. On Page 3, Line 158, the authors describe equation (6) as a "third-order tensor method." It would be more accurate to characterize it as a second-order method with third-order regularization, since the subproblem in (6) involves only up to second-order derivatives.

4. Minor typo: On Page 26, Line 1352, in the first equation, $\nabla^2 f(x - \tau(y - x))$ should be corrected to $\nabla^2 f(x + \tau(y - x))$.

**Questions:**

See the weakness.

---

> ### Author Response · Authors · 2025-11-26
> **Rebuttal**
>
> We thank Reviewer tDN2 for the careful, in-depth reading and many specific comments that will help us improve both clarity and positioning. We are happy to address any further questions during the discussion.
>
> ### RN stepsize schedule vs AICN stepschedule
>
> We agree that our original phrasing was imprecise. Under the same semi-strong self-concordance assumptions, RN does **not** reproduce the AICN stepsize exactly. What it does share is the **asymptotic dependence** on both the gradient norm and the smoothness constant. We will revise the text to clearly state this asymptotic relationship rather than suggesting an exact match.
>
> ### Assumption 1 and Hanzely et al. (2022)
>
> It is correct that Assumption 1 does not appear explicitly in Hanzely et al. (2022). We intended to refer to a related condition used there, namely
> $$
> \nabla^2 f(x_{k+1})^{-1} \preceq (1-c) \nabla^2 f(x_k)^{-1},
> $$
> which appears in the proof of local quadratic convergence (Lemma 4, equation (25)). We will rephrase our statement to reflect this more accurately.
>
> ### Characterizing equation (6)
>
> We agree with the reviewer that equation (6) is more appropriately described as a *second-order method under third-order similarity*, rather than a “third-order tensor method.” We will adopt this terminology.

---

> > ### Author Response · Authors · 2025-12-01
> > **Response follow-up**
> >
> > We sincerely appreciate your time and would be thankful if you could examine both our rebuttal response and the revised version of the manuscript, and, if they address your points, kindly reconsider the current score.

---

### Official Review · Reviewer_jsfd · 2025-10-31

**Soundness:** 3
**Presentation:** 3
**Contribution:** 2
**Rating:** 6
**Confidence:** 4

**Summary:**

This paper revisits the global convergence properties of **Newton-type methods** under general **Hölder continuity** assumptions on the Hessian (or higher derivatives).
Building on the **AICN framework of Hanzely et al. (2022)**, the authors propose a family of **stepsize rules** derived from solving a polynomial equation that relates gradient and Hessian norms.
This formulation can be interpreted as a *third-order extension* of damped or regularized Newton methods, leading to improved global convergence rates up to **O(k⁻³)** without requiring explicit Hölder constants.
The paper also introduces adaptive variants (GRLS and UN) that use backtracking or linesearch strategies to estimate stepsizes in practice. Experiments on convex and mildly nonconvex problems support the theoretical findings.

**Strengths:**

- **Solid theoretical foundation:** The work extends and generalizes the ideas in AICN to broader Hölder-smooth settings.
- **Interesting parametrization:** The introduction of the parameter \( \theta \) to implicitly define the stepsize \( \alpha \) is mathematically elegant.
- **Adaptive schemes:** The GRLS and UN variants are practical contributions that make the theory more broadly applicable, removing dependence on unknown Hölder constants.

**Weaknesses:**

- **Dependence on prior work:** Much of the structure and analysis builds directly on **Hanzely et al. (2022)**, with limited new conceptual ideas beyond the general Hölder extension.
- **Clarity issues:**
  - The role of the **parametrization \($\theta$ \)** and how it determines the stepsize \( $\alpha$ \) is not clearly explained. This deserves more detailed discussion, since it is central to the contribution.
  - The main problem is that $\theta$ depends on $\alpha$ but in the algorithm it does not.
- **Adaptive method overhead:** The adaptive (backtracking) procedures may add nontrivial computational cost. The paper would benefit from an explicit discussion or empirical measurement of this overhead.
- **Scope limitation:** The analysis and guarantees are developed specifically for **convex deterministic functions**, and it is unclear how (or whether) these results extend to **non-convex** or **stochastic** optimization settings.
- **Experimental scope:** The experiments are relatively small and primarily illustrative. Additional comparisons with standard baselines such as L-BFGS or trust-region Newton methods would strengthen the empirical section.

**Questions:**

1. Could the authors clarify **how the dependence between \( \theta \) and \( \alpha \)** was handled in both analysis and implementation?
   - Is \( \theta \) treated as an implicit function of \( \alpha \), or is it estimated independently?
   - How sensitive is the method to errors in estimating or initializing \( \theta \)?
2. The results are presented for **convex deterministic** objectives.
   - How might the proposed analysis or algorithms extend to **non-convex** settings, where the Hessian might not be positive definite?
   - Could the same framework apply to **stochastic** optimization, where gradients and Hessians are noisy?
3. In adaptive variants (GRLS and UN), what is the **practical computational overhead** of the backtracking procedure, and how does it scale with problem dimension?  I believe you should be able to have a theoretical guarantee for the backtracking approach.

---

> ### Author Response · Authors · 2025-11-26
> **Rebuttal**
>
> ## Response to Reviewer jsfd
>
> We thank Reviewer jsfd for the thoughtful and constructive feedback, and for highlighting several points where additional clarification will improve the paper. We are happy to answer any follow-up questions.
>
> ### Dependence on prior work
>
> Our analysis builds on the AICN framework of Hanzely et al. (2022), but extends it to general Hölder smoothness, and to adaptive stepsizes strategies. We will adjust the wording to better reflect this and avoid overstating conceptual novelty.
>
> ### Dependence between $\theta$ and $\alpha$
>
> For a given iterate $x_k$, $\theta_k$ and $\alpha_k$ are in one-to-one correspondence via
> $$
> 1-\alpha_k - \alpha_k^{1+\beta}\sigma\Vert \nabla f(x^k)\Vert _{x^k}^{*\beta}=0
> $$
> (with fixed $\beta$ and $\sigma$). Specifying either $\theta_k$ or $\alpha_k$ uniquely determines the other, and every admissible $\theta_k$ corresponds to a valid $\alpha_k$.
>
> A key observation is that the one-step decrease condition in Theorem 2 can be expressed entirely in terms of $\theta$. Hence, implementing a Newton method with stepsize $\alpha_k$ corresponding to the $\theta_k$ from Theorem 2 directly yields the desired decrease. Intuitively, choosing a larger $\theta$ can be viewed as fixing a larger effective regularization parameter $\sigma$.
>
> ### Sensitivity in estimating/initializing $\theta$
>
> We do not estimate $\theta$ as an external parameter. Instead, $\theta_k$ is an implicit parametrization of the chosen stepsize $\alpha_k$; there is no separate estimation procedure whose errors we need to control. If the reviewer had a different interpretation in mind, we would be happy to clarify further in the revision or discussion.
>
> ### Cost of the adaptive procedures (GRLS and UN)
>
> - **GRLS line search.**
> Linesearch GRLS is not intended as a practical algorithm. We propose it for 3 reasons
>
>   (i) demonstrate the meaningfulness of the one-step decrease formula,
>   (ii) motivate the practical UN backtracking procedure, and
>   (iii) connect our analysis to classical Greedy Newton line search.
>
> - **UN backtracking.**
>   Naively implemented, one backtracking step has roughly the same cost as one Newton step. As each backtracked round increases the effective $\sigma_k$ by a constant factor, the number of backtracked rounds is at most $\log(\sigma)$, as we show formally in Lemma 3. We will emphasize this more clearly in the text.
>
> ### Beyond convex deterministic functions
>
> Our work is focused on stepsizes for Newton methods under Hölder continuity of third derivatives. This is largely orthogonal to stochasticity and nonconvexity, but we see promising directions:
>
> - **Nonconvex settings.**
>   Shestakov et al. (2025) show that damped Newton stepsizes are loss-transformation invariant and that many pseudoconvex objectives can be transformed to convex ones. This relaxes convexity to pseudoconvexity in a natural way.
>
> - **Stochastic settings.**
>   There exist many stochastic Newton-type schemes; our stepsize design is compatible with several of them, and exploring this rigorously is part of our ongoing research.
>
> ### Comparisons with L-BFGS and trust-region methods
>
> We will clarify our experimental design:
>
> - When Hessians are computable and Newton steps are viable, quasi-Newton methods (including L-BFGS) expected to be much slower.
> - When Hessian computation is infeasible (large-scale settings), Newton methods are not competitive, and L-BFGS or first-order methods are the natural choice.
> - Trust-region methods are primarily motivated by nonconvex settings. For convex problems where Newton is viable, trust-region constraints offer limited benefit.
>
> Given these structural differences, we believe including L-BFGS or trust-region baselines would not add much insight into the behavior of the step-size strategies we study. We would be happy to discuss alternative experimental setups if the reviewer has specific suggestions.

---

> > ### Author Response · Authors · 2025-12-01
> > **Response follow-up**
> >
> > Dear reviewer,
> > We would greatly appreciate it if you could revisit our rebuttal response and the updated paper, and, if the changes satisfactorily resolve your concerns, consider updating your score.

---

### Official Review · Reviewer_2vmp · 2025-11-06

**Soundness:** 3
**Presentation:** 2
**Contribution:** 2
**Rating:** 4
**Confidence:** 3

**Summary:**

In this paper, the authors study the damped Newton's method for convex functions whose Hessians or third derivatives are Hölder continuous. The key idea is to apply higher-order regularization to Newton's method in the local norm induced by the Hessian, which is shown to be equivalent to using a specific damped step size. Under the assumption of bounded Hessian variation, they establish a suplinear local rate and a global sublinear rate up to $O(k^{-3})$. Since the derived step size depends on the unknown Hölder continuity exponent and constant, they further propose both exact and backtracking line search schemes that achieve the same global rate without prior knowledge of these parameters.

**Strengths:**

The paper builds on Doikov et al. (2024), who proposed Newton's method with higher-order regularization in $\ell_2$-norm and proved an $O(k^{-3})$ rate when the third derivative is Lipschitz continuous. However, that approach sacrifices affine invariance due to the use of $\ell_2$-norm. To address this limitation, similar to Hanzely et al. (2022), the authors adopt regularization in the local norm, thereby restoring affine invariance while retaining the same accelerated rate under higher-order smoothness. The convergence analysis is thorough and includes line-search variants that eliminate the need for prior knowledge of problem parameters.

**Weaknesses:**

- My main concern lies in Assumption 1, which appears somewhat restrictive.
   - The authors cite Hanzely et al. (2022) for this assumption. However, in my understanding, their global sublinear rate does not rely on such an assumption, and a similar condition is only used to establish local quadratic convergence near the solution. In contrast, both the global and local convergence results in this paper depend on this assumption.
  - The authors argued that this assumption is satisfied when the function is $L$-smooth and $\mu$-strongly convex, or when it has a stable Hessian. Yet, in both cases, one would typically expect a faster global linear convergence rate, rather than the slower sublinear rate obtained here.
  - The authors also noted that this assumption holds for self-concordant functions when iterates are close to each other or in the neighborhood of the solution. However, it remains unclear whether this condition is applicable for global analysis, since the iterates may move far in a single step.
- I think the prior work by Doikov et al. (2024) is highly relevant and deserves a more detailed discussion in the main text. Specifically, their regularized Newton method achieves a comparable rate of $O(k^{q-1})$ under (the more standard) smoothness assumption in the $\ell_2$-norm. It will also be helpful if the authors can elaborate on how their proof techniques differ from those in that work.

**Questions:**

I find the proofs in the appendix somewhat difficult to follow and suggest that the authors improve their presentation. For example, in the proof of Lemma 4 for the case $p=3$, it is unclear to me how the inequality in line 1527 is derived.  I assume Lemma 1 is used in this step, but additional clarification would be helpful.

---

> ### Author Response · Authors · 2025-11-26
> **Rebuttal**
>
> ## Response to Reviewer 2vmp
>
> We thank Reviewer 2vmp for the detailed, thoughtful review and the many helpful suggestions. We are happy to elaborate further on any of the points below.
>
> ### Assumption 1 and its restrictiveness
>
> We agree with the reviewer, but would like to point out that we stated the Assumption 1 in a more restrictive fashion that we actually use. Our global convergence uses Assumption 1 only in inequality (120), so we don’t need this bound to hold uniformly at every iteration – it suffices to have *geometric average* of those “local” $\gamma_k$ values to be well behaved. This is weaker than demanding a single global $\gamma$ valid for all steps. We omitted this viewpoint for brevity and will clarify it in the revision.
>
> ### Relation to Doikov et al. (2024) and differences in proof techniques
>
> We agree that Doikov et al. (2024) deserves a more detailed discussion. Due to space constraints, we placed most of it in Appendix A; we will move a more explicit comparison to the main text.
>
> **Technical differences:**
>
> - **Local norms and unknown Hölder exponent.**
>   The proofs of Doikov et al. (2024) cannot be directly adapted to local norms (as in Hanzely et al. 2022). In our case, there appears the stepsize $\alpha_k$ raised to the power $1+\beta$, which propagates nontrivially throughout the analysis and complicates direct adaptation.
>
> - **Implicit reparametrization via $\theta$.**
>   Our key insight is the reparametrization (line 141), where a single parameter $\theta$ encapsulates both $\beta$ and $\sigma$. This reparametrization allows us to recover a proof structure similar to Doikov et al. (2024) while avoiding direct manipulations of $\alpha_k^{1+\beta}$.
>
> - **Affine-invariant framework.**
>   Working in local norms lets us leverage identity (4) to relate gradient norms and model decreases in the same geometry, leading to tighter estimates in some steps. While Doikov et al. (2024) focus on $\ell_2$ norms, local norms also appear throughout their proofs. We avoid non-affine-invariant norms entirely, which also makes some arguments (e.g., bounds on third derivatives) simpler and more interpretable.
>
> ### Clarification of the inequality in line 1527
>
> The inequality follows from two applications of the Cauchy–Schwarz inequality. Firstly
> $$
> -\langle a,b\rangle \ge -\Vert a\Vert \cdot \Vert b\Vert  \ge -\tfrac12\Vert a\Vert ^2 - \tfrac12\Vert b\Vert ^2,
> $$
> with
> $a=\nabla^2 f(x^k)^{-1/2}\nabla f(x^{k+1})$,
> $b=\nabla^2 f(x^k)^{-1/2}\nabla^3 f(x^{k})[x^{k+1}-x]$, and secondly
> $$
> (1-\alpha_k)\langle a,b\rangle \ge -(1-\alpha_k)\Vert a\Vert \cdot \Vert b\Vert ,
> $$
> with
> $a=\nabla^2 f(x^k)^{-1/2}\nabla f(x^{k})$,
> $b=\nabla^2 f(x^k)^{-1/2}\nabla^3 f(x^{k})[x^{k+1}-x]$.

---

> > ### Author Response · Authors · 2025-12-01
> > **Response follow-up**
> >
> > Dear reviewer,
> > We would be very grateful if you could kindly review our rebuttal response and the revised version of the paper, and, if you feel that your concerns have been adequately addressed, consider adjusting the score.

---

### Author Response · Authors · 2025-11-26
**Rebuttal comment**

We thank all reviewers for their careful reading and constructive feedback. Below we address the comments point by point and will incorporate the suggested clarifications in the camera-ready version. We are happy to further clarify any of these points. We will revise the manuscript to:

- clarify assumptions and notation,
- soften and correct novelty claims where needed,
- shift the discussion of most related works to the main paper body,
- improve the explanation of RN/UN/GRLS and their practical roles,
- and make the technical arguments and proof steps easier to follow.

We are grateful for the opportunity to improve the paper and are happy to answer any additional questions or concerns.

---

### Meta-Review · Area_Chair_QNQr · 2026-01-06

**Summary:**

This paper proposed Newton-type methods with the global convergence rate of $O(1/k^3)$ for minimizing convex function with Hölder continuous Hessians or third derivative. The main contribution is the new stepsize schedule and the universal stepsize backtracking for known problem parameters. I think the content of this paper is above the borderline.

**Reviewer Concerns:**

The main questions:
1. Assumption 1 is restrictive.
2. Comparison with Doikov et al. (2024).
3. The dependency between $\theta$ and $\alpha$.
4. The experiments only report the convergence, rather than the matrix operations.

I think the main concerns have been addressed.
1. Assumption 1 holds for many cases, such as the smooth and strongly convex objective.
2. Compared with Doikov et al. (2024), the technical differences of this paper include the local norm, addressing the case of unknown Hölder exponent, implicit reparameterization, and affine-invariant framework.
3. This has been explained in rebuttal.
4. Figures 2 and 3 report the convergence in wall-clock time.
Based on the reviewers’ comments and rebuttal, I tend to accept this paper. However, the rebuttal revision uses the NeurIPS 2025 template. I'm not sure whether the author used the correct template in the initial submission. In this situation, should the paper be desk rejected?

**Reviewer Scores:**

Reviewer 2vmp may raise the score, since the authors have explained the concerns on Assumption 1.

---

### Decision · Program_Chairs · 2026-01-26

Accept (Poster)